

# 1 Thermal-Driven Graupel Generation Process to Explain Dry-Season Convective Vigor over the Amazon

**Authors**: Toshi Matsui[1&2], Daniel Hernandez-Deckers[3], Scott E. Giangrande[4], Thiago S. Biscaro[5], Ann Fridlind[6], and Scott Braun[1]

[1] *Mesoscale Atmospheric Processes Laboratory, NASA Goddard Space Flight Center, Greenbelt, MD, USA*

[2] *Earth System Science Interdisciplinary Center – ESSIC, University of Maryland, College Park, MD, USA*

[3] *Grupo de Investigación en Ciencias Atmosféricas, Departamento de Geociencias, Universidad Nacional de Colombia, Bogotá, Colombia*

[4] *Environmental and Climate Sciences Department, Brookhaven National Laboratory, Upton, NY, USA*

[5] *Meteorological Satellites and Sensors Division, National Institute for Space Research, Cachoeira Paulista, São Paulo, Brazil*

[6] *NASA Goddard Institute for Space Studies, New York, NY, USA*

**Correspondece to:** Toshi Matsui, Toshihisa.Matsui-1@nasa.gov





**Abstract**. Large-eddy simulations (LESs) are conducted for each day of the intensive observation
periods (IOPs) of the Green Ocean Amazon (GoAmazon) field campaign to characterize the
updrafts and microphysics within deep convective cores while contrasting those properties
between Amazon wet and dry seasons. Mean Doppler velocity ($V_{dop}$) simulated using LESs are
compared with 2-year measurements from a Radar Wind Profiler (RWP) as viewed by statistical
composites separated according to wet and dry season conditions. In the observed RWP and
simulated LES $V_{dop}$ composites, we find more intense low-level updraft velocity, vigorous graupel
generation, and intense surface rain during the dry periods than the wet periods. To investigate
coupled updraft-microphysical processes further, single-day golden cases are selected from the
wet and dry periods to conduct detailed cumulus thermal tracking analysis. Tracking analysis
reveals that simulated dry-season environments generate more droplet-loaded low-level thermals
than wet-season environments. This tendency correlates with seasonal contrasts in buoyancy and
vertical moisture advection profiles in large-scale forcing. Employing a normalized time series of
mean thermal microphysics, the simulated cumulus thermals appear to be the primary generator of
cloud droplets. At the same time, ice crystals tend to be generated in inactive parts of clouds. Time
series shows that thermals, however, entrain ice crystals and enhance riming due to large
concentrations of droplets in the thermal core. This appears to be a production pathway of
graupel/hail particles within simulated deep convective cores. In addition, less-diluted dry-case
thermals tend to be elevated higher, and graupel grows further during sedimentation after spilling
out from thermals. Therefore, greater concentrations of low-level moist thermals likely result in
more graupel/hail production and associated dry-season convective vigor.



## 1. Introduction

Deep convection is a fundamental process of turbulence that drives the Earth's general circulation
and regulates thermodynamic fields (Emanuel et al., 1994). Deep convection undergoes complex
dynamical and microphysical processes throughout its life cycle, which appear as towering clouds
visible from satellites in different parts of the world (Stephens et al., 2002). As a result, deep
convection generates significant amounts of atmospheric latent heat, surface precipitation, and
hydrometeors that reflect/absorb solar and infrared radiation, modifying atmospheric circulation
and surface energy and mass fluxes (Hartmann, 2016). These complexities in deep convection and
feedback processes pose significant challenges in predicting weather and climate using numerical
Earth system modeling across different scales (Grabowski and Petch, 2009; Sullivan and Voigt,

56  2021).


Characteristics of deep convection are unique in different seasons and geographic regimes affected
by the local environment. One of the most straightforward yet most robust regime separation
concepts is the land-ocean (L-O) contrast (Williams and Stanfill 2002). Solar radiation warms the
surface skin temperature over land more readily than over the ocean due to the smaller heat
capacity of soils and vegetation than deep water bodies, thus producing stronger surface infrared
flux and turbulent heat flux (Matsui and Mocko 2014). This greater surface energy deepens
planetary boundary layers that may trigger deeper convective clouds depending on the atmospheric
profiles (Pielke 2001). Overall, the continental environment tends to promote deeper convection
with stronger, wider convective cores (Lucas et al. 1994, Wang et al. 2019), with suppressed warm
rain and enhanced cold precipitation process (Williams et al. 2005), which often leads to unique
drop-size distribution characteristics and precipitation partitioning between convective and
stratiform process outcomes in different geographic regions (e.g., Tokay and Short 1996;
Giangrande et al. 2012; Dolan et al. 2018).

Satellite observations similarly depict continental convective invigoration as characterized by
more frequent lightning flashes and heavily rimed particles aloft over land than ocean (Williams
et al. 2004, Zipser et al. 2006, Stolz et al. 2015, Matsui et al. 2016). Takahashi et al. (2017 & 2021)



show that continental convection generally contains less diluted cores than their oceanic
counterparts, following an inverse relationship between convective core width and dilution rate.
Similarly, Jeyaratnam et al. (2020) recently suggested that convective updraft and mass flux
properties were distinctly different between tropical land and tropical oceanic convection using
methods to estimate those properties that blend satellite observations with plume models.
Hereafter, we define "convective vigor" by the enhanced cold-precipitation process characterized
by larger rimed particles (graupel/hail) and vigorous raindrops in convective cores.

Representation of deep convective cloud land-ocean contrasts is still an ongoing challenge for
global atmospheric models at storm-resolving resolution (a few km of horizontal grid spacing),
partially owing to the poor representation of cloud dynamics (Matsui et al. 2016). Robinson et al.
(2011) configured idealized simulation setups to investigate the "island" effect of convection with
finer grid spacing (500 m or 1 km) and successfully simulated convective vigor equivalent to the
observed microwave brightness temperature. Matsui et al. (2020) used a nested regional model
with 1 km grid spacing to compare mid-latitude continental versus tropical maritime storms and
successfully reproduced land-ocean contrasts of hydrometeor identification profiles from
polarimetric radars.

However, statistical evaluation of simulated vertical velocity and association with convective
vigor process (i.e., graupel/hail generation) have not yet been examined very well due to a lack of
observations and detailed process-oriented model investigation, respectively. For example,
mesoscale convective system (MCS) studies performed by Prein et al. (2022) and Ramos-Valle et
al. (2023) highlight the challenges when attempting to represent continental convection within the
constraints of limited observations while attempting to establish optimal configurations as a
function of model grid spacing for typical midlatitude (Oklahoma) and tropical continental
(Amazon) conditions.

The 2-year measurements from the Green Ocean Amazon (GoAmazon) campaign provide
unprecedented data on the vertical velocity of deep convection by the Atmospheric Radiation
Measurement user facility (ARM, ARM Mobile Facility (AMF), Martin et al. 2017, Giangrande
et al. 2017). Recently, Giangrande et al. (2023) contrasted the thermodynamics and lifecycle





properties, including the vertical air velocity within isolated convective clouds observed during
the Amazon wet and dry seasons, and found that dry-season convection exhibited more intense
low-level updrafts and stronger precipitation properties associated with smaller convective cell
areas than wet-season counterparts. Dry-season convection also tended to exhibit a  shorter life
cycle and often achieved maximum updraft and precipitation intensity at earlier life cycle stages
than wet-season storms. Pre-convective thermodynamics profiles from those events revealed that
the dry season showed a stronger deficit of dew-point temperature in the middle troposphere and
higher values of mean-layer convective available potential energy (MLCAPE) at lower levels.

In addition to primary thermodynamic transitions between wet and dry season convective regimes
(e.g., Giangrande et. al. 2020), the Amazon dry seasons may experience larger concentrations of
aerosols due to biomass burning that have been recently associated with potential secondary
contributions to changes in storm precipitation properties and convective vigor (e.g., Lin et al.
2006; Wang et al. 2018; Öktem et al. 2023). Moreover, the Amazon dry (wet) season has long
been suggested to promote a continental (maritime) convection contrast for a given
thermodynamic profile and background aerosols (Williams et al. 2002). Typical land-ocean
contrast is characterized by a "hot" continental surface (Williams and Stanfill 2002) and sea-breeze
type of mesoscale dynamics due to the thermal-patch effect (Robinson et al. 2011). Thus, instead
of focusing on the complex nature of land-ocean contrast or other active versus break monsoonal
contrasts performed globally (e.g., Holland et al. 1986, Keenan and Carbone 1992, Pope et al.
2009, Wu et al. 2009), the dry-wet season contrasts over the Amazon basin allows a unique
emphasis on the impact of thermodynamic profiles and large-scale dynamics upon the formulation
of convective vigor.

The main objective of this paper is to investigate dry-wet seasonal contrast and potential changes
in the evolution of deep convection cloud processes using an LES model and forward simulations
of Doppler radar observations. This attempts to reveal dynamical and microphysical processes that
explain the observed dry-wet contrasts, focusing on the bulk controls imposed by the background
thermodynamic profiles and large-scale forcing. The motivation for these efforts is the argument
that an improved understanding of these dry-wet contrasts should facilitate untangling the more
complex processes of land-ocean contrasts in deep convection. For this study, we employ a series





of daily large-eddy simulations (LESs) with bulk single-moment microphysics throughout the
GoAmazon campaign dry and wet season intensive observation periods (IOPs) to characterize dry-
wet season contrast in convection. These simulations are validated against the cumulative statistics
collected by ground-based Doppler velocity measurements during those periods. A thermal
tracking analysis is conducted to select golden cases from these dry- and wet-season LES runs to
investigate the physical process of convective vigor further. This effort focuses on thermodynamic
impacts and cloud dynamical roles in this manuscript, while the aerosol effects will be investigated
in another study.

Section 2 describes the general methodology and tools, including radar profilers, LES,
instrumental simulators, large-scale meteorological forcing, and thermal-tracking algorithms for
this study. Section 3 shows the results of the dry-wet contrast of meteorological forcing, statistical
composites of Doppler velocity, and thermal tracking analysis. Section 4 summarizes a thermal-
driven process of dry-season convective vigor, its uncertainties, and future directions.


**2.        Methods**

**2.1 GoAmazon: RWP Observations**

The primary datasets for this study are those collected by the 1290 MHz ARM Radar Wind Profiler
(RWP) operated during the U.S. Department of Energy's ARM ARM facility deployment during
its "Observations and Modeling of the Green Ocean Amazon 2014–2015" (GoAmazon) campaign
near Manaus, Brazil, from March 2014 through December 2015 (e.g., Martin et al., 2017). The
RWP was configured for precipitation sampling that included frequent vertical pointing to collect
conventional radar reflectivity factor Z and mean $V_{dop}$ profiles through deep convective cells to
approximately 17 km in altitude (e.g., Giangrande et al. 2013, Giangrande et al. 2016, Wang et al.
2019, Williams et al. 2023). For measurements collected at these ultra-high frequencies and
without expectations for larger hail in Amazon deep convective storms, all radar estimates are
assumed as within Rayleigh scattering regimes, and measurements are unattenuated in rain. The
RWP deployed during GoAmazon had a beam width of approximately 10 degrees thus horizontal



measurement resolution is typically less than 1 km, with a 200 m vertical (bin, gate) resolution and
10-s intervals between consecutive radar profiles. All radar measurements were calibrated against
a reference laser disdrometer collocated at the main AMF site during the campaign (e.g., Wang et
al. 2018).

For observation and simulation comparisons, the deep convective core is defined by using
thresholds applied to the observed and simulated RWP profiles: 1) column-maximum reflectivity
is greater than 35 dBZ, and 2) column-maximum $V_{dop}$ is greater than 5 m/s.  The choice for these
criteria is admittedly flexible, as model-vs-observed Z thresholds, in particular, are not necessarily
well-posed for convective-stratiform segregation such as removing all stratiform cells (e.g., Steiner
et al. 1995). Still, the additional velocity constraint afforded by the vertically-pointing RWP and
statistical representation of convective and stratiform composites seems to be reasonable (not
shown here), noting what is important is an attempt to apply thresholds for both observations and
simulations. A slightly different threshold does not alter the conclusions. Note that events on
3/19/14, 3/20/14, and 3/23/14, as well as the 10/4/14 cases from the dry season IOP, are clear
examples of Amazon mesoscale convective systems (MCSs, e.g., Wang et al. 2019). Thus, these
cases have also been removed from our statistical analysis to focus on isolated convective days.


**2.2 GoAmazon LES, Forcing, and Simulator**

GoAmazon LES runs utilize the Goddard Cumulus Ensemble (GCE), a cloud-process model
developed and improved at NASA GSFC over several decades (Tao *et al.* 2014).  The GCE is
driven by large-scale forcing (LSF) with cyclic boundary conditions to generate cloud dynamics
and microphysics processes in Cartesian grid coordinates. No additional heat, moisture, or
momentum enters the domain apart from that imposed by the LSF or solar/infrared radiative
processes. In addition, GCE's anelastic dynamic core option allows faster integration of finer-
resolution runs (up to 1.5~2 times) than its compressible dynamic core option.

GoAmazon LES runs use 200-m horizontal grid spacing with 512 x 64 x 128 grids (x-y-z cartesian
coordinates) with a 2-second model time step. Vertical grid spacings stretch from near surface



level (starting from 44 m) and reach 200m around 4 km level, not to exceed horizontal resolution.
Thus, the domain covers a 102 km x 12.8 km area; this "narrow channel" domain setup intends to
resolve three-dimensional large eddies (i.e., thermals) while minimizing computational cost in
order to run LES for the entire IOPs. In terms of model physics, the 1.5-order turbulent kinetic
energy (TKE) scheme is used for subgrid turbulent mixing, and the Goddard radiation scheme is
used for computing radiative flux and heating (Chou and Suarez 1999 & 2001, Matsui et al. 2018).
The Goddard bulk one-moment 6-class scheme (4ICE hereafter) has four ice classes and uses
preset size and density mapping for snow, graupel, and hail (Lang et al. 2014; Tao et al. 2016).
4ICE successfully generated a realistic L-O contrast of convective-core hydrometeor distributions
compared to polarimetric radar retrievals in the previous study (Matsui et al. 2020). Also, note that
the one-moment scheme is unaffected by the background aerosol concentrations to focus on the
impact of thermodynamic and large-scale forcing on convective vigor in this study.

The LSF is derived from the VARiational ANALysis (VARANAL) approach, which is a broadly
accepted method for generating large-scale forcing wherein data are collected and adjusted based
on the vertical integration of the atmospheric mass, moisture, dry static energy, and momentum
budgets (Zhang and Lin 1997, Zhang et al. 2001, Xie et al. 2004). The VARANAL approach is
applied to the GoAmazon field campaign using ERA-Interim reanalysis (Dee et al., 2011) and
constrained by radar-based surface precipitation rate (Tang et al. 2016). GoAmazon LESs are run
from September 2014 to 10 October 2014, defined as the dry-season IOP, and also run from 14
February 2014 to 26 March 2014, defined as wet-season IOP, as suggested by thermodynamic
behaviors characteristic of larger dry and wet-season expectations, respectively (Giangrande et al.
2017, 2020). Each daily LES is initialized at 12 AM local time and integrated just for 30 hours
rather than continuously integrated during the entire IOPs because the convection life cycle
typically follows a strong diurnal cycle due to the solar heating cycle, excepting propagating
organized convection (Tang et al. 2016, Giangrande et al. 2017, 2020). As a default setting, hourly
LES outputs are used to analyze the mean seasonal behavior of LESs.

Hourly LES outputs include an additional Doppler velocity field, corresponding to an expected
RWP observation through a multi-instrumental simulator, Goddard Satellite Data Simulator Unit
(G-SDSU, Matsui et al. 2014a; Matsui et al. 2014b). In this study, a ground-based Doppler radar



simulator is implemented in the model to replicate RWP observable signals. Radar backscatter is
estimated from nonRayleigh calculations with a Maxwell-Garnett assumption of air-water-ice
mixtures at 1290 MHz frequency, though for the RWP frequency, vertical pointing, and media
type/size expectations therein, the forward modeling is more straightforward than most weather
radar wavelength applications and appropriate for Rayleigh scattering assumptions. Doppler
velocity is estimated using pressure-adjusted hydrometeor terminal velocities weighted by radar
backscatter spectrum for each particle size distribution (PSDs). All these single scattering
calculations follow the 4ICE microphysics calculation/assumptions of particle size, density, and
phase for each hydrometeor species for physics consistency (Matsui et al. 2014b). Finally,
simulated signals are averaged consistent with the RWP beamwidth (10 degrees). This beamwidth
implies different averaging at different heights, for example, corresponding to six horizontal grids
of the LESs being averaged for a representative output at the 15 km height. Overall, this beamwidth
averaging smear LES-scale Doppler velocity signal statistics closer to the anticipated observed
instrumental signals (Matsui et al. 2014b). However, it should be noted that the exact sampling
methods are different between observations and simulations. For example, the RWP observations
are vertical pointing measurements that collect profiles at 6-second ("instantaneous") intervals. In
contrast, the modeled RWP signals are drawn from a domain-wide sampling of hourly LES
outputs.


**2.3 Thermal Tracking Algorithm**

The thermal tracking method used here is described in detail by Hernandez-Deckers and Sherwood
(2016), who improved the initial version used by Sherwood et al. (2013). It is an offline algorithm
that uses high temporal resolution output (~1 min) from LES to identify and track coherent rising
volumes of cloudy air, i.e., thermals. The algorithm first identifies all peak vertical velocities larger
than 0.8 m/s that have water condensate content of at least 0.01 g/kg at every available snapshot
of the simulation and matches peaks from successive snapshots to identify the available points of
the trajectories of rising cloudy air parcels. A third-order polynomial is fitted to these points to
reconstruct smooth trajectories and to allow a precise estimate of the ascent rate of the rising air
volume at each snapshot. Notice that this ascent rate differs from the actual vertical velocity at a



particular grid point since thermals develop internal toroidal circulations such that the peak vertical
velocity at their centers is higher than the actual ascent rate of the air volume (e.g., Blyth et al.
2005; Sherwood et al. 2013). The extent of each rising air volume (the size of each thermal) is
estimated assuming a spherical shape centered at its smoothed trajectory, ensuring that the average
vertical velocity of the enclosed volume matches that obtained from the derivative of the trajectory.
Tracked thermals must fulfill certain requirements; for example, they must be tracked for at least
three time steps, their radius must be larger than twice the horizontal grid spacing, their time-
average ascent rate must be at least 1 m/s, their change in size in between successive snapshots
must be less than 80% of the smallest radius, and most importantly, their trajectories must be
consistent with their vertical momentum budget. The momentum budget of a tracked thermal is
computed from its buoyancy (obtained from the density field), the pressure gradient force
(obtained by integrating the pressure field over the entire thermal's surface), a "resolved mixing
term" (obtained from the convergence of vertical momentum flux across the thermal's surface),
and an entrainment or detrainment contribution due to the change in size between snapshots. This
allows us to compute the expected final position of each thermal based only on its initial ascent
rate, which is compared with the thermal's last tracked position. The distance between the actual
and expected final positions must be smaller than the average thermal diameter and smaller than
20 % of the vertical distance traveled; otherwise, the thermal is discarded. Once thermals are
tracked with this algorithm, many properties can be studied based on all available model variables
of interest. For example, average values for each thermal, such as ascent rate, size, altitude,
entrainment rate, etc., are easily computed. Also, composites of different quantities can be obtained
for different "stages" of a thermal's lifecycle. Typically, thermals exhibit one maximum ascent rate
throughout their lifetime, which indicates their most vigorous phase. This time step is used as a
time reference common (t=0) to all thermals to create composites of various properties at different
stages of thermal lifetimes.

This thermal tracking algorithm was first used to study the main properties of cumulus thermals in
simulations of transient-growing convection (Sherwood et al., 2013; Hernandez-Deckers and
Sherwood, 2016) and provided strong evidence that thermals are typically small, short-lived (4-5
minutes on average), and mix vigorously with their environment. Also, Hernandez-Deckers and
Sherwood (2016) showed that the spherical shape approximation is generally valid and that



thermals, rather than plumes, are a more realistic building block for cumulus clouds. Hernandez-
Deckers and Sherwood (2018) used this algorithm to study the mixing properties of thermals in
more detail and contrast them with known parameterizations. Results from these studies have set
up the stage for a deeper understanding of cumulus dynamics and for further studies that use
different approaches (e.g., Gu et al., 2020; Morrison et al., 2020; Peters et al., 2020; Xu et al.,
2021; Morrison et al., 2023). Recently Hernandez-Deckers et al. (2022) used this algorithm to
study aerosol-deep convection interactions, highlighting the importance of the strong coupling
between microphysics and small-scale dynamics in convective clouds. Here we run this tracking
algorithm with the GCE model output for 5 hours starting at 1900Z (3 pm local time) on 09/07/14
(dry case) and 02/26/14 (wet case), using 1-minute interval output.


**3. Results**


**3.1 Dry-Wet Contrast of Large-Scale Forcing**

Campaign atmospheric thermodynamic profiles and the typical variability observed during
GoAmazon dry and wet seasons have been previously depicted using composite radiosonde skew-
T log-P diagrams (e.g., Giangrande et al., 2017, 2020, 2023). These depictions often show very
similar temperature profiles between dry and wet seasons, whereas the moisture profiles indicate
apparent differences, highlighting the mid-level deficit of the dew-point temperature profile in dry-
season composites. Since this study utilizes LSF to drive LESs, seasonal thermodynamics and
dynamics are re-characterized by the LSF (Tang et al. 2016).

In Fig. 1, we plot a time series of apparent moisture sinks (Q2), vertical moisture advection, and
parcel potential buoyancy profiles with surface precipitation rate from GoAmazon LSF for the
IOPs. These time series of LSF profiles are integrated and contrasted in terms of Contoured
Frequency by Altitude Diagrams (CFADs, Yuter and Houze 1995) as the dry and wet season IOPs
(Figure 2).

Here, Q2 is the sum of changes in moisture content, horizontal moisture advection, and vertical
moisture advection (Yanai et al. 1973), balanced with net condensation rate and turbulent transport



of moisture vertical advection. Large Q2 corresponds to a large atmospheric moisture loss due to
net condensation loss (i.e., precipitation). Large Q2 is associated with intervals with heavier or
more widespread surface precipitation; thus, dry-IOP Q2 and surface precipitation are typically
smaller than wet IOP (Fig. 1a-b). Similarly, Figs. 1c-d shows that peaks of vertical moisture
advection term coincide with those peaks in the Q2 rate. Note that the Q2 rate in tropical
environments is mainly contributed by the vertical moisture advection term rather than the
horizontal advection term (not shown here). More importantly, positive (red shade) vertical
moisture advection of the wet IOP tends to be stretched up to higher altitude (up to 200 mb) than
the dry IOP (up to 600 mb) in most cases.

As previously discussed by Tang et al. (2016), the associated Amazon Q2 CFADs show the largest
positive Q2 between 700 and 400 mb, while the largest negative Q2 is around 800 mb (Figs. 2a-
b). The Dry-wet composite CFAD highlights more frequent positive Q2 values above the 800 mb
level during the wet IOP. In contrast, more frequent negative Q2 during the dry IOP (Fig. 2c).
Vertical moisture advection depicts similar CFAD shapes (Figs. 2d-e). Still, it highlights high
frequencies of low-level positive vertical moisture advection and mid-to-low-level negative
moisture vertical advection in the dry IOP in comparison with the wet IOP.

Finally, in Figs. 1e-f we plot the time series of parcel potential buoyancy profiles (positive
components only), computed from LSFs by lifting surface airmass dry and moist adiabatically.
These potential buoyancy magnitudes are not necessarily associated with precipitation intensity.
Potential buoyancy CFADs show peak forcing between the 600 mb and 200 mb levels (Figs. 2g,
h, & i). The wet IOP suggests a larger variability of potential buoyancy at the upper troposphere
than the dry IOP (Figs. 2g-h). Potential buoyancy appears to be slightly stronger in the dry IOP,
and concentrated in a relatively lower troposphere than its wet IOP counterpart (Fig. 2i), which
agrees with findings in Giangrande et al. (2023). These results will be further discussed along the
thermal concentrations in the latter section.


**3.2 Dry-Wet Composite of Doppler Velocity CFADs**

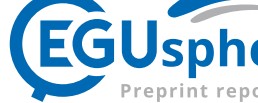

Giangrande et al. (2023) highlighted dry-wet seasonal characteristics of storm vertical air motions
retrieved using RWP. They found that daytime isolated dry season convective cells tend to have
stronger updrafts at altitudes below the melting level. Yet, unlike their wet-season counterparts,
updrafts do not increase in intensity much above the melting layer. However, dry-season
convective cores were also characterized by stronger downdrafts at all altitudes, especially when
compared to wet-season counterparts aloft. Our present study utilizes similar direct measurements
of the mean $V_{dop}$ from RWP to characterize the dry-wet contrast of deep convective cores. The
advantage of using $V_{dop}$ measurements is that the quantity is the direct radar measurement and
helps reduce uncertainties from retrieval assumptions, such as requiring hydrometeor
identification or associated terminal fall speed corrections if the intent was to retrieve the vertical
air motion (Giangrande et al. 2013, 2016). Here, vertically-pointing $V_{dop}$ measurements contain
sufficient information to evaluate storm characteristics, with the understanding that these
measurements represent the terminal velocities of hydrometeors combined with the vertical air
motion.

In Fig. 3a, we provide the cumulative sample numbers of CFADs (for each bin of $V_{dop}$ and altitude)
as simulated and subsampled from the LES hourly outputs from the combined dry and wet season
IOPs. If the sampling numbers are normalized for each altitude, this will form the $V_{dop}$ CFADs to
follow. Fig. 3b shows the sum of hydrometeor mass concentrations from each $V_{dop}$-altitude bin.
Namely, each hydrometeor mass concentrations from "cloud", "rain", "graupel-plus-hail", or "ice-
plus-snow" are separately accumulated for each bin. The larger number of samples associated with
a larger accumulated mass concentration of hydrometeor can generate the "representativeness" of
the hydrometeor class for a given $V_{dop}$-altitude bin location.

As mentioned above, we defined four regimes based on the accumulated mass of each hydrometeor
category. The "cloud" category (CL) is centered around -5 m/s of $V_{dop}$ and 4 km altitude, slightly
overlapping with other categories. A "rain" category (RA) is more narrowly concentrated around
-8 m/s of $V_{dop}$ and below 4 km altitude. The "graupel-plus-hail" category (GH) is centered around
-14 m/s of $V_{dop}$ at 5 km altitude. Finally, an "ice-plus-snow" category (IS) is narrowly concentrated
along -1 m/s of $V_{dop}$ above 5 km altitude. These locations roughly correspond to each hydrometeor
category's altitude and terminal velocity when factoring in the background/ambient vertical air





velocity. Note that our "cloud" regime has no terminal velocity in GCE 4ICE microphysics, thus
$V_{dop}$ represents or tracks the background vertical air velocity and overlaps with the other regimes.
Moreover, simulated $V_{dop}$ and hydrometeor statistics are also sensitive to model physics and those
assumptions to some degree. For example, any real-world cloud regime may be extended to higher
altitudes, but the model 4ICE microphysics scheme tends to quickly convert cloud liquid to cloud
ice category due to saturation adjustment (See Figs. 14-16 of Matsui et al. 2023). Nevertheless,
this representative mapping will help discuss the variability of the $V_{dop}$ CFADs between the dry
and wet season IOPs.

In Fig. 4, we provide an observed and simulated climatology of $V_{dop}$ CFADs as sampled from deep
convective cores and summarized over the dry and wet season IOPs. In both the dry and wet season
IOPs, the observed CFADs depict a smoother transition of the $V_{dop}$ at the freezing level into the
melting layer (4-5 km, Figs. 4a-b). At the same time, simulations show a more abrupt transition
around the freezing layer (Figs. 4d-e). This is primarily because bulk single-moment microphysics
more abruptly converts solid to liquid phases through autoconversion than explicit bin-resolving
microphysics (Iguchi et al. 2014). This rapid conversion also overestimates the terminal velocity
of raindrops near and just below the freezing level.

The CFADs have been summarized according to dry and wet season IOPs to explore these seasonal
contrasts between the deep convective cores (Figs. 4c, 4f). In the R regime (green box), the dry
IOP suggests more prevalent samples in strongly negative $V_{dop}$ for our observations and
simulations, indicating that deep convective cores during the dry season IOP tend to have more
vigorous, faster-falling (larger) raindrops. In the GH regime (purple box), the dry season IOP
dominates the sampling. The observations indicate this dominance (red shade) up to 10 km altitude
(the extent that observations were included), while the simulation shows this behavior up to 8 km,
suggesting LES underestimation in graupel/hail altitudes. In the CL regime, the observations and
the simulations agree well, except that some sampling is overwhelmed by the dry season IOP
behaviors in the overlapped area. This likely indicates a shift in the presence of stronger low-level
updraft velocities, consistent with the analysis in Giangrande et al. (2023).



When considering the IS regime, there are examples of agreements and discrepancies between the observations and simulations. One key agreement is that the wet IOP dominates the samples in the area of positive $V_{dop}$ above 8 km altitude. This indicates that observations and simulations suggest a shift towards stronger upper-level vertical air velocity for the wet season IOP examples than for the dry season IOP. As before, this is consistent with the absence of dry mid-levels and the stochastic updraft model expectations from Giangrande et al. (2023). On the other hand, the observations indicate a more dominant sampling of velocities during the wet season IOP at around -3 m/s of $V_{dop}$, whereas simulations change the dominant sampling mode from wet to dry IOPs at around 7 km altitude. This is a potential bias in single-moment bulk microphysics, which tends to glaciate cloud droplets or raindrops more quickly into ice particles than double-moment schemes (e.g., Fig. 16 of Matsui et al. 2023). The observed composite also shows more dry-season dominant frequencies in GH zones than the simulation, indicating the underestimation (overestimation) of raindrop/graupel (ice/aggregate) particles above 7 km height.

Excepting this discrepancy in the IS regime, dry-wet composites of $V_{dop}$ CFAD agree well between observations and simulations, suggesting that LES could successfully represent the important nature of dry-wet contrast, i.e., dry-season convection tends to generate stronger low-level updraft velocity, generating more graupel/hail, and vigorous raindrops accompanied with stronger low-level downdraft than the wet season.

To further investigate these seasonal shifts in core properties, golden cases are selected to analyze deep convection lifecycle and processes. Namely, we select two single-day simulation cases representing typical dry and wet-season convection. For this, the $V_{dop}$ CFADs are constructed for each day during the wet and dry season IOPs, and these daily CFADs are compared to the composites of seasonal CFADs (not shown here). After day-to-day analysis of correlation and root-mean-square errors between daily and seasonal CFADs, the convective event on the 2/26/14 case is selected to represent the wet IOP convections. In contrast, the 9/7/14 case is chosen to represent dry IOP convection. Fig. 5 shows a dry-wet composite of $V_{dop}$ CFAD using these two case studies. This figure compares quite well with the seasonal composite of $V_{dop}$ CFAD, having the dry convective vigor signals and model biases of the seasonal composite (Fig. 4f).



In Fig. 6, we show a time series of domain-mean profiles of convective cores drawn from these
dry- and wet-season golden cases, highlighting from 1600Z (1200 local time) of the starting day
to 0400Z (0000 local time) of the next day. The dry season golden event shows a clear diurnal
convection cycle, peaking at 2100-2200Z (local 5-6 pm). In contrast, the wet season golden event
shows an already ongoing, continuous sequence of deep convection with several embedded strong
pulses. Convective top heights reach up to 17 km for both the dry and wet events (Figs. 6a-b).
Low-level positive upward vertical velocity is more ubiquitous in the dry case, while upward
vertical velocity of the wet case extends to the middle-to-upper troposphere up to 15km (Figs. 6c-
d).

Dry-case graupel-plus-hail (GH) mass concentrations peak around 2100-2300Z when the
convective clouds reach their deepest cloud top heights, and the maximum GH concentration
exceeds that of the wet case. Rain mass concentrations peak between 2200Z and 00:30Z on the
subsequent day for the dry case, and this appears to be slightly more intense than the rainfall
simulated for the wet cases. Note that precipitation areal fraction is expected to be larger for the
wet season (i.e., Giangrande et al. 2016, 2023), such that dry-season convection is often
characterized by narrow yet intense isolated convection, while wet-season convection is
characterized by widespread moderate to deep convection (although with higher domain mass
flux). The intense surface rainfall rates are generally correlated with the generation of graupel,
frozen drops, and/or small hail particles during the dry-season convection, but there are some time
lags from 21Z to 22:50Z in the dry-case convection. This is because the initial convective core is
much narrower, and near-surface relative humidity is slightly low (~80%) around 21Z, and later
convective area increases so as near-surface relative humidity (~96%) around 21:50Z. Thus, more
surface rain evaporation likely suppresses surface precipitation during earlier convective periods.
These time series behaviors are generally consistent with the observed characteristics in the time-
integrated $V_{dop}$ CFADs (Fig. 5).

One key question is why larger or heavily rimed particles tend to be preferentially generated in
dry-season convection compared to wet-season convection, given that both seasons indicate
convection with intense updraft velocity. This question follows previous efforts of Williams and
Stanfill (2002) for simulations of deep convection that contrasted land and oceanic clouds. For





example, while land and ocean environments may have similar convective available potential
energy (CAPE), differences in detailed potential buoyancy and vertical velocity profiles enable
additional graupel/hail particles to form in continental deep convection when compared to the
maritime environments (Matsui et al. 2020). A Lagrangian tracking analysis is performed to
examine this question for Amazon dry and wet season contrasts to investigate the dynamics and
microphysics within cumulus thermals for these dry and wet golden events (Section 3.3).


**3.3 Thermal Tracking Analysis**

Thermal tracking analysis (Section 2.3) was conducted over 5-hour periods from 1900Z to 0000Z
for the dry and wet season events using 1-minute LES outputs. Fig. 7 depicts normalized x-z cross-
sections of thermal properties at the moment of maximum vertical velocity in the dry and wet cases
and dry-wet differences. Thermals typically experience development and decaying stages in their
lifetime. During development, moist thermals increase their vertical velocity and size by releasing
latent heat and entraining surrounding air (Morrison et al. 2021). After defining and tracking each
thermal from the LESs, our normalizing procedure first defines the reference time from each
thermal's lifetime based upon peak vertical air velocity (denoted as thermal maxima) and then
conducts a weighting average of each thermal property at the thermal maxima only. Our weights
are based on the magnitude of thermal mass flux to avoid under-representing properties of less-
populated but vigorous thermals. Because these heights at thermal reference time are different for
each thermal in dry and wet case studies, averaging properties are somewhat biased toward thermal
vertical distributions  (discussed later).

For example, in Fig. 7a we plot the weighted-average peak vertical air velocity (W) for the dry-
case thermal (9.6 m/s) and the wet-case thermal (10.6 m/s). Perhaps surprisingly, the flux- and
radius-weighted average dry-case thermal is slightly slower in W than that found for the average
wet-case. Here, we find that the vertical air velocity of the wet-case thermal is more
homogeneously distributed than its counterpart for the dry-case thermal, leading to higher
weighted-mean W despite weaker values at thermal centers (red shade in Dry-Wet plot, Fig. 7a).
Also, unexpectedly, supersaturation and cloud droplet mixing ratio ($Q_c$) of the dry-case thermal



are elevated compared to the wet-case thermal (Fig. 6b-c), since wet-case thermals may be
expected to instead have higher supersaturation and/or more condensation owing to the higher
availability of water vapor (e.g., Giangrande et al. 2023).

Exploring the other classes, the rain mixing ratio (Qr) is similar between the dry-case and wet-case
thermals (0.17 g/kg), but graupel-hail concentrations (Qg+h) are significantly larger in the dry-
case thermals (0.95 g/kg) compared to the wet-case thermals (0.79 g/kg); this latter result is
consistent with previous discussions from event time-series in Fig. 6e. Cloud ice and snow mixing
ratio (Qi+s) values are slightly larger in the wet-case thermal (3.5 g/kg) than in the dry-case thermal
(3.2 g/kg). While this difference is not significant, this is also potentially a surprising outcome
since dry-case deep convective clouds might otherwise be expected to be deeper/stronger and thus
characterized by additional ice hydrometeor concentrations. However, some absence of these
media may be partially explained by following Giangrande et al. (2020; 2023) suggestions that
drier mid-to-upper levels in the dry season may limit periphery precipitation aloft (i.e., enhanced
evaporation). Overall, Qr and Qg+h seem to be concentrated in these composite averages
downward from the thermal core due to the gravitational sedimentation process. Supersaturation
and Qc, however, are also more vertically elongated than thermal properties established by
Hernandez-Deckers et al. (2022) using the Weather Research and Forecasting (WRF) model for a
case of scattered convection over Houston, Texas. Qi+s is more homogeneously distributed across
the defined borders of thermals. Also, dry-wet differences show slight asymmetric results,
particularly in W, Qr, and Qg+h. These could be attributable to differences in horizontal wind
shear, evidenced by a greater tilt in the thermal centerline flow in the dry case (gray streamlines),
leading to greater concentrations in the tilt direction of more rapidly sedimenting quantities that
are formed within thermals (i.e., Qr and Qg+h); since thermal composites are not aligned with the
mean wind, such preferential outflow may not be fully captured by this analysis (i.e., asymmetric
signatures could be greater or lesser along other directions than X alignment).

An initial leading question is why the dry-case thermals have greater cloud water and
supersaturation on average. To further untangle these results in Fig. 7b-c, we derive the vertical
profiles of flux-weighted mean thermal states, now including all thermal times (Fig. 8a-f).
Immediately, these plots reveal striking differences between the thermal number concentration



(N) profiles for dry-case and wet-case examples (Fig. 8a; the number of thermals per km height
within the 102 km x 12.8 km domain). For instance, dry-case convection shows a larger
concentration of thermals below the 8 km height, while wet-case convection promotes a more
homogeneous thermal concentration that extends across most heights. This behavior is somewhat
reminiscent of the distribution for the difference in vertical moisture advection and potential
buoyancy profiles between the parent dry and wet season conditions (e.g., Fig. 1 and 2, discussions
in Section 3.1). Moreover, thermal generation in our LES responds to these terms partially from
the seasonal large-scale forcing.

According to the classic similarity theory of Morton et al. (1956), the width of thermals should
increase with increases in the boundary layer depth (William and Stanhill 2002). For the Amazon
basin, previous GoAmazon studies such as Giangrande et al. (2017, 2023) showed that dry season
boundary layer height is generally deeper than that of the wet season, potentially on the order of
200 m deeper for isolated deep convective events they tracked. Following this logic, dry-case
convection may anticipate larger thermals. However, LES thermal tracking analysis suggests that
the sizes (R) of thermals upon initiation appear to be quite similar between the wet and dry events
and then appear to grow at similar rates for several km before the dry-case thermal size catches up
with the moist size around 6 km in height, only to be overtaken again by the deeper wet-case
thermals around 9 km (Fig. 7b). This result implies that differences in moist convection between
dry and wet cases are perhaps better characterized by thermal numbers rather than thermal sizes.

Qc in thermals also shows very similar profiles between the dry and wet cases (Fig. 8c). However,
because thermal numbers of the dry case are more concentrated at the lower troposphere (Fig. 8a),
all-height mean properties of dry-case thermals are characterized by more Qc (Fig. 7c). Qr of the
wet case is nearly twice as large as the dry case (Fig. 8d); however, normalized x-z cross-section
(Fig. 7d) does not show such a large difference (explained below). Qi+s also shows similar
distributions (Fig. 8e). Still, total x-z mean Qi+s is larger in the wet case than the dry case due to
larger thermal numbers in the upper troposphere (Fig. 7e). Uniqueness appears in thermal Qg+h
(Fig. 8f). While both dry and wet cases show similar magnitude of the peak values (~0.9 g/kg), the
peak height in the dry case is approximately 3 km higher than the wet case.





Fig. 8g-j displays these hydrometeor mixing ratios averaged over the same periods, including all
convective grids defined by vertical velocity greater than 1 m/s. Vertical profiles and dry-wet
differences are similar to the results in Fig. 6. However, compared with the in-thermal profile
results (Fig. 8c-f), it facilitates understanding of the convective core microphysics process. First,
mean in-thermal convective-grid hydrometeor concentrations are smaller than in-thermal profiles;
particularly in-thermal Qc values are roughly six times larger than convective-grid average Qc
(Figs. 8c & 8g), suggesting that thermals are major cloud droplet generators (Hernandez-Deckers
et al. 2022).

The convective-grid Qg+h of the dry case is nearly twice as high as that in the wet case, peaked
around the melting layer (Figs. 8f & 8j), whereas in-thermal Qg+h shows similar peak values
between the dry and wet cases. As indicated by Fig. 7f, these larger and heavier rimed particles
sediment from thermals and further collision with ice and supercooled liquid must enhance the
graupel growth during the sedimentation process, as suggested from aircraft measurements (Blyth
and Latham 1993). Thus, elevated in-thermal Qg+h in dry-case convection can have further riming
growth after falling out from thermals. This vigorous growth of Qg+h in dry-case convection
eventually generates vigorous raindrops after the melting process. This is why convective-grid Qr
in the dry case is larger than that in the wet case (Fig. 8h), opposite from the result of in-thermal
Qr (Fig. 8d). Thus, in-thermal Qr values are not directly related to total Qr in the convective core
(or surface precipitation rate) because of this cold precipitation microphysics process in deep
convection.

Now, a second leading question is why the height at the peak value of dry-case in-thermal Qg+h
is more elevated than the wet-case thermal (Fig. 8f). Fig. 9 shows histograms of thermal properties
from the dry and wet cases. Consistent with the mean vertical profiles (Fig. 8a), more thermals are
initiated below 7 km in the dry case than in the wet case (Z0, Fig. 9d). Thermal radius in the wet
case is also larger than the dry case regardless of shallower boundary layer depths in the wet case
(Fig. 9a), consistent with R in thermal vertical profiles reaching larger sizes at most elevations in
the wet case (Fig. 8b). However, here we see that thermal vertical velocity (W, Fig. 9b), travel
distance (dZ, Fig. 9c), and lifetime (Fig. 9e) in the dry case are all greater than in the wet case.
Thermal entrainment rate is smaller in the dry case than the wet case. These results indicate that



the thermals in the dry-case deep convection can travel longer distances with an extended lifetime
due to a lesser dilution.

Giangrande et al. (2022) suggest that the convective area is smaller in dry-season convection over
this region. Thus, this indicates that stronger low-level buoyancy in dry-season environments can
more narrowly concentrate updraft and low-level thermals in the area, thus creating less diluted
environments probably due to the impact of thermal drag (Romps and Charn 2015). Takahashi et
al. (2022) investigated cloud-scale entrainment between continental and maritime environments
and found a larger dilution rate in maritime convection than in continental convection. Our results
suggest that this difference in cloud dilution happens from the thermal-process level. These
conditions elevate dry-case thermals and graupel peak concentration toward higher altitudes than
the wet-case convection (Fig. 7f), leading to greater graupel production.

Finally, time series of thermal properties in the x-z cross-section are constructed for the dry case.
For this, each thermal at its maximum w value is centered and defined as the time of zero, and
prior (later) steps are represented in negative (positive) time steps. Because of the 1-minute LES
output, the time series from -3 to 3 encompasses 7 minutes of time steps. This averaging process
also weighs upon the magnitude of the thermal mass flux (Hernandez-Deckers et al. 2022); thus,
thermals at larger values in positive and negative time steps tend to have lesser sampling numbers.
Also, to make the composites, equal-sized thermals are sampled to characterize the mean time
series of thermal properties, avoiding sampling too small thermals, which often has no
supersaturation (Hernandez-Deckers and Sherwood. 2016). This normalizing procedure ends up
with the result that maximum W values do not appear at reference time (t=0), but better capture
the evolution of the largest flux-bearing thermals (Fig. 10). We also note that a typical thermal
travel distance is 1.3 km (Fig. 9c) and a minority of dry-case thermals therefore contain either no
ice phase (Fig. 8e-f) or no liquid phase (Fig. 8c-d), but most contain both phases between 3 and 7
km. Note that this flux-weighting is the one way to present the results, while simple non-weighting
averaging can also show similar results.

In the dry case (Fig. 10), within thermals that experience an extended peak in W (6–11 m s$^{-1}$), the
average supersaturation, cloud, and rain mixing ratio peaks at the earlier steps and decreases





toward the end of the time steps. This indicates that a chunk of condensation heating is the main
initial driver of moist thermal growth. These thermal properties are typically centered around the
thermal core. By contrast, Qi+s properties are more homogeneous and less concentrated at the core
of thermals, and they tend to increase toward the end of the time series. Especially, the early stages
(t=-3, -2, & -1) indicate thermals are approaching an existing ice layer rather than generating ice
around the thermal core. In the later stages (t=1, 2, & 3), the Qi+s is weakly concentrated toward
the upper thermal cores. This evolution of Qi+s suggests that thermals are not the main initiator of
Qi+s, while Qi+s is rather entrained into the thermal within the early stages of the mixed-phase
zone, at least using the single-moment bulk microphysics. On the other hand, after liquid saturation
is no longer contributing substantially to Qc, Qi+s becomes a leading destination of the overall
transfer from vapor to hydrometeor phases within thermals that remain vigorous. This also
suggests that the glaciation process (i.e., conversion from supercooled liquid to ice hydrometeors)
is usually completed after thermals vanish unless they reach the upper level of convective cores.

On the other hand, Qg+h increases toward the peak time of thermals (t=0), and starts decreasing
toward the later time steps (t=3). The spatial concentration of Qg+h is also peaked around the
thermal cores, similar to W, rh, Qc, and Qr. The increase of Qg+h coincided with the timing of
thermal entrapment of Qi+s and a reduction in Qc and Qr for time steps between -3 and 0. This
suggests that large concentrations of in-thermal Qc and Qr collide with entrained Qi+s to enhance
the riming process, generating graupel and hail particles. After the reference time step (t=0), Qg+h
decreases, most likely due to sedimentation exceeding production. As indicated by Fig. 8f & 8j,
this spilled graupel and hail can further grow by colliding with supercooled liquid and ice particles
until melting. Taken together, this analysis also suggests that this vigorous Qg+h-generation
process in the convective core *does not* occur through the classic parcel-driven convection; i.e., a
large single air mass lifted from the cloud base up to the cloud top can generate latent heat and
precipitation (Arakawa and Schubert 1974). Instead, these graupel and hail generations are most
likely driven by sequential interactions of thermal ensembles and microphysical processes. Note
that the time series of the wet case also shows a similar finding but is biased toward the thermals
in the upper atmosphere (not shown here).






**4. Conclusion: Thermal-driven Convection Invigoration Process**

We have investigated seasonal differences of the measured and simulated $V_{dop}$ between the dry and wet seasons to confirm dry-season convective vigor associated with enhanced cold precipitation (graupel and hail) processes. Tracked thermal properties from the selected case studies reveal unique updraft microphysics processes in the convective core that explain the dry-wet contrast in deep convection. To summarize our findings in graupel-hail development sequences, a thermal-driven process is proposed in the following steps (Fig. 11a).

1. Where condensation may occur within moist turbulent structures in the lower atmosphere, local moist thermals may be initiated, which are characterized by dipole vortex rings with intense vertical velocity, supersaturation, cloud droplets, and raindrops around the thermal core.
2. When moist thermals penetrate the 0°C isothermal layer and dissipate in the mixed-phase zone, cloud droplets are detrained and gradually glaciated to form ice-particle layers.
3. As additional thermals fill with droplets and penetrate the glaciated mixed-phase zone, they entrain ice particles and collide with each other, generating graupel and hail embryos.
4. Once graupel and hail particles grow sufficiently large, they start falling out from thermals and develop further by collecting supercooled droplets and ice particles during sedimentation.

The process of generating ice layers (Step 3) could be the largest source of uncertainty in this study. To prove the convective vigor process, this study used the simple bulk single-moment microphysics parameterization (Lang et al. 2014, Tao et al. 2016). This parameterization tends to convert droplets into ice particles through the saturation adjustment process. Cloud droplets are glaciated much more quickly when compared to two-moment microphysics (e.g. Matsui et al. 2023). Time series data also shows some ice generation near the thermal core in later lifecycle stages, which may be associated with homogeneous freezing, vapor deposition or riming. Yet, ice crystal formation processes remain one of the largest sources of microphysics uncertainty (Kanji et al. 2017; Korolev and Leisner 2020) and need further investigation to establish and adequately parameterize. Furthermore, updrafts passing through the melting layer containing both large drops





and ice crystals (which are identified here as a source of graupel) have also been pinpointed as a
leading source of secondary ice production in oceanic convection sampled extensively via aircraft
(Korolev et al. 2020). Thus, all quantitative components of the proposed ice-graupel generation
process described here remain uncertain and subject to future investigations via instrumental
observations and more detailed numerical simulations.

Nonetheless, building on the ability of existing knowledge and simulations to reproduce some
basic features of observations during GoAmazon, Fig. 11b shows a newly proposed process that
can explain why dry-season convection has more graupel and intense precipitation than wet-season
counterparts in the following steps.

1. Dry-case (wet-case) convection tends to generate more (less) numbers of droplet-loaded

thermals from the lower atmosphere because of larger potential buoyant energy at a low

level in the dry season.

2. Once an ice layer is built from initial cumulus thermal ensembles (Fig. 11a), more (less)

numbers of droplet-loaded thermals penetrate ice layers to generate more (less) graupel and

hail embryos in dry-case (wet-case) convection.

3. Individual dry-case (wet-case) thermals can rise to higher (lower) elevations via weaker

(stronger) dilution, elevating in-thermal graupel at higher (lower) altitudes.

4. During sedimentation, graupel in dry-case (wet-case) thermals has a higher (lower) chance

to grow due to the longer (shorter) distance toward the melting level.


The "hotter" surface in the dry season must be the physical origin of step 1, similar to L-O contrast
(William and Stanfill, 2002). The dry season typically has clearer skies, less soil moisture, and
stronger surface heating, leading to more turbulent heat flux and energy at the lower troposphere
even during GoAmazon experiment (Biscaro et al., 2021; Ghate and Kollias, 2016). In contrast,
the wet season is characterized by frequent precipitation and increased release of atmospheric
latent heat with the weak surface sensible heat flux caused by wet soil moisture (Rocha et al.,
2004). As a result, the entire troposphere experiences upward motion during the wet season, unlike
its dry season counterpart (Tang et al., 2016).



Contrary to the speculation made by William and Stanfill (2002), it has been found that stronger
surface heating and deeper PBL during the dry season do not increase the thermal "size" based on
the classic similarity theory of Morton et al. (1956). Instead, our analysis of simulations indicates
that the "numbers" of cumulus thermals become more important, particularly those initiated in the
lower troposphere. Even for similar CAPE, the concentration of potential buoyancy profiles in the
lower troposphere can trigger more vigorous convection. This is similar to the difference between
mid-latitude continental and tropical maritime environments, where the mid-latitude continental
environment tends to have more potential buoyancy in the mid-to-lower troposphere, leading to
continental convective vigor (Matsui et al., 2020).

It is also hypothesized that the low-altitude concentrated cumulus thermal trains could additionally
enhance the cold precipitation process by improving the residence time of graupel and hail within
the mixed-phase zone if thermal-spilled graupel and hail encounter subsequent new cumulus
thermals. Previous trajectory modeling (Heymsfield 1983) also suggested a similar mechanism for
enhancing graupel and hail residence time and growth by multiple convective cores. Heymsfield
(1983) used the multi-Doppler technique to generate a three-dimensional wind field, but it needed
more spatio-temporal resolution to characterize cumulus thermal. However, a stronger updraft core
mentioned in his study must be cumulus thermals. This investigation further requires a more
complex set of numerical simulations in the future.

The proposed thermal-driven invigoration process is based solely on thermodynamics and does
not consider aerosols' effect on deep convection, as demonstrated by previous studies over the
Amazon (William et al. 2002, Lin et al. 2006). Our choice of single-moment microphysics does
not consider the variability of background aerosols to initiate cloud droplets. However, this simple
microphysics can generate a fundamental dry-wet contrast characterized by the $V_{dop}$ statistics. This
suggests that thermodynamics is the primary factor determining convective vigor, while aerosols
may have a significant but secondary role in invigorating convection (Matsui et al. 2020). Future
studies will require a higher-order moment of microphysics scheme to examine the impact of
aerosols on droplet and primary ice nucleation in thermals to confirm our hypothesis that dry-wet
aerosol contrast plays a secondary role.



The proposed process for graupel-hail generation and convective vigor is a time-dependent,
sequential, coupled dynamics-microphysics process. This process is not linear and cannot be
adequately represented by the traditional convective mass flux method (Arakawa and Schubert
1974). To represent this process, thermal chain concepts with detailed microphysics processes
must be introduced in the parameterization for large-scale models (Morrison et al., 2020). Fine-
resolution simulations produce better continental convective vigor because they can resolve
thermal dynamics and microphysics (Robinson et al., 2011; Matsui et al., 2020). The mean radius
of the tracked thermal in this study, conducted using a 200 m mesh LES, is around 1 km, with a
maximum size of around 2 km, which is comparable to the LES study using a 65-m horizontal grid
spacing (Hernandez-Deckers and Sherwood, 2016). However, due to the effective resolution being
5-10 times the actual grid spacing, cumulus thermals, and graupel-hail generation processes are
difficult to resolve for storm-resolving models and perhaps any Eulerian-type numerical
atmospheric models (Matsui et al., 2016). Conducting LES for regional and global weather and
climate models is impractical in the foreseeable future. Therefore, new types of dynamics-
microphysics-coupled cumulus thermal parameterization should be developed to better represent
deep convection for storm-resolving and coarse-resolution weather and climate models.

New ground-based Doppler phased array radar (PAR) technology (Kollias et al. 2022b) or multi-
Doppler agile scans (Kollias et al. 2022a) hold promise in observing and characterizing cumulus
thermals. Emerging PAR instruments have started capturing storm motion and microphysical
details at spatial and temporal resolutions akin to those seen in LES output (e.g. Takahashi et al.
2019, Kikuchi et al. 2020). These new observational capabilities are necessary for refining the
dynamics and microphysics in LESs, particularly in elucidating the process behind thermal-driven
convective vigor. Moreover, the advent of vertical motion estimates from high-resolution space-
based radars [EarthCARE, Wehr et al. 2023; Investigation of Convective Updrafts (INCUS),
https://incus.colostate.edu; the Atmosphere Observing System (AOS), https://aos.gsfc.nasa.gov)
will soon enable the global mapping of convective updrafts. These new satellite radar
measurements will generate a comprehensive global catalog detailing convective vigor and the
speed of intense thermals.



**Code Availability.** The GCE LES code, G-SDSU simulator code, and Python plotting codes used in this mansucript are all available in the NCCS Data Portal (https://portal.nccs.nasa.gov/datashare/cloudlibrary/PUB_DATA/GoAmazon_ACP/Code/).

**Data Availability.** The RWP measurements and VARNAL LSF data were available from the Atmospheric Radiation Measurement (ARM) ARM Data Discovery (https://adc.arm.gov/discovery/#/). These data were obtained from the ARM Mobile Facility (AMF) at Manacapuru, Amazonas, Brazil, funded by A. U.S. Department of Energy (DOE) Office of Science User Facility managed by the Biological and Environmental Research program. The analysis data used in this manuscript is also available in the NCCS Data Portal (https://portal.nccs.nasa.gov/datashare/cloudlibrary/PUB_DATA/GoAmazon_ACP/Data/)

**Author contribution.** T. Matsui designed and performed the GCE LESs, the $V_{dop}$ forward simulation, and the thermal tracking. D. Hernandez-Deckers developed the thermal tracking and analysis code for the GCE LESs and prepared the $V_{dop}$ figures for the thermal analysis. S. Giangrande and T. Biscaro prepared RWP $V_{dop}$ analysis. T. Matsui prepared the manuscript with contributions from all co-authors.

**Competing interests.** At least one of the (co-)authors is a member of the editorial board of Atmospheric Chemistry and Physics.

**Acknowledgments.** This project is funded by the NASA CloudSat and CALIPSO Science Team (CCST) program (program manager: Dr. David Considine, grant number: 80NSSC21K1135). The development of GCE is funded by the NASA GSFC Strategic Science resources (Associate Director for Institutional Planning and Operation: Dr. Karen Mohr). We also thank the NASA Advanced Supercomputing (NAS) Division and Center for Climate Simulation (NCCS) (Project Manager T. Lee at NASA HQ) for providing the computational resources to conduct the GCE and G-SDSU simulations, thermal tracking analysis, and stored model outputs. The authors thank an anonymous reviewer for improving the manuscript.



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



**Figures**

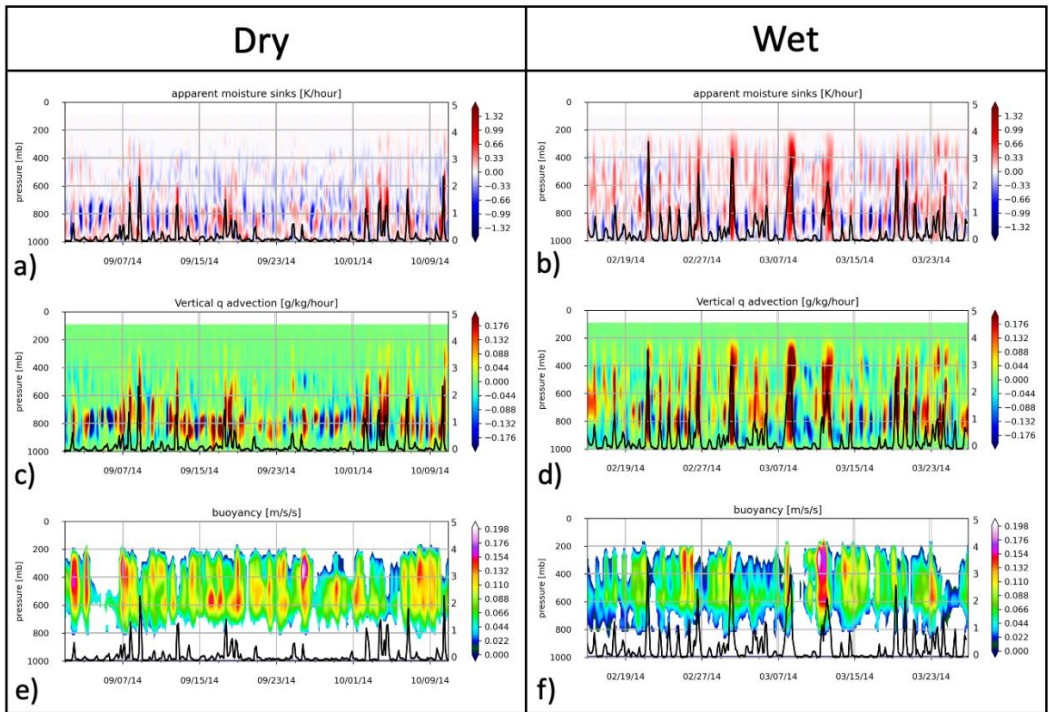


Figure 1. Time series of VARNAL large-scale forcing profiles between wet and dry periods: (a-b)
apparent moisture sink (Q2), (c-d) vertical moisture advection, (e-f) potential buoyancy. The black
solid lines on the secondary y-axis represent the surface precipitation rate.



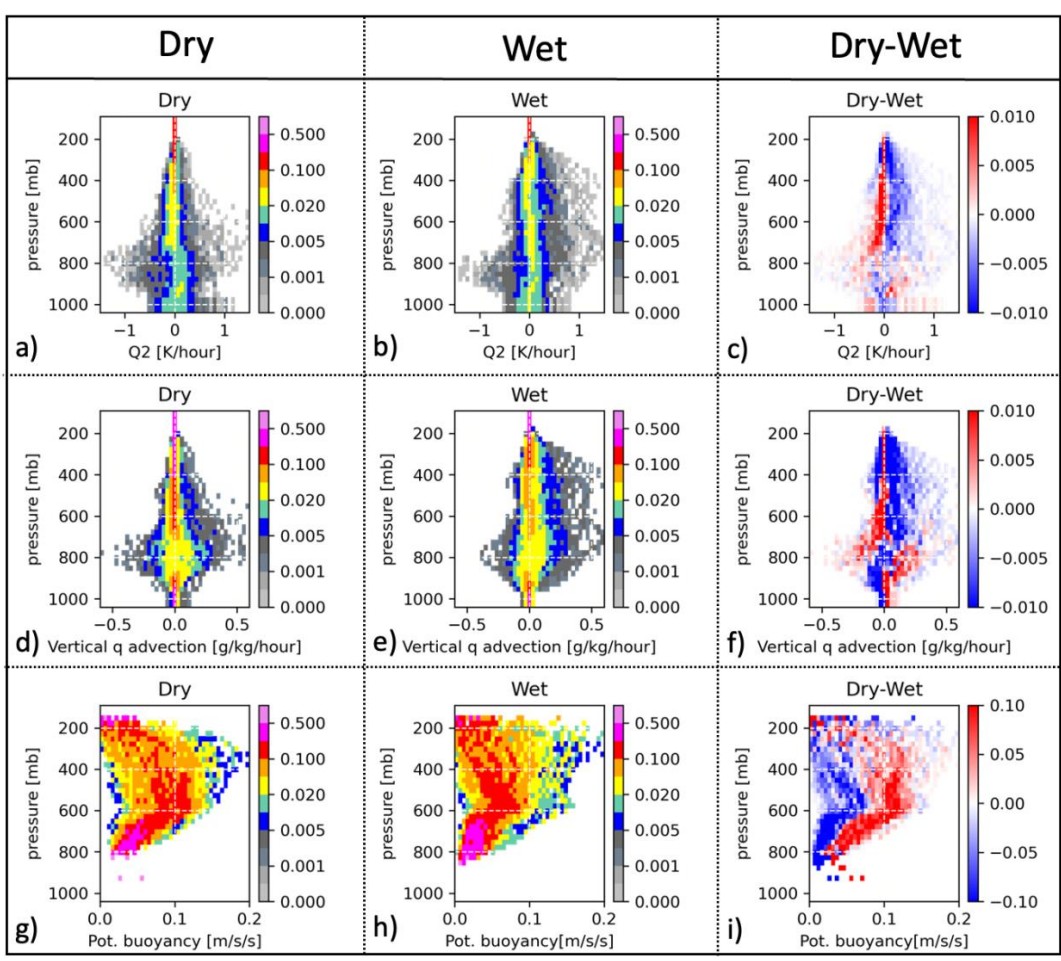


Figure 2. Contoured Frequency of Altitude Diagram (CFADs) of (a-c) apparent moisture sink
(Q2), (d-f) vertical moisture (q) advection, (g-i) potential buoyancy integrated over dry and wet
periods, as well as dry-wet differences.



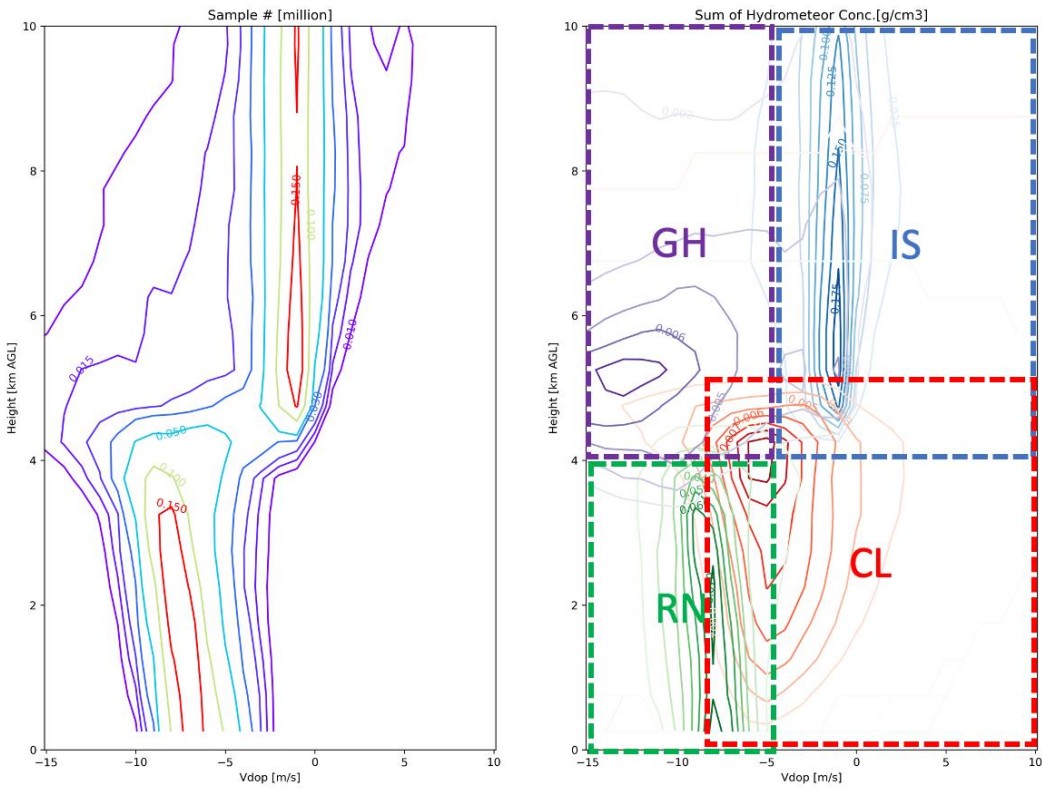

Figure 3. (a) Cumulative $V_{dop}$ sample numbers from LESs during dry and wet periods, presented as CFADs for each $V_{dop}$ bin and altitude. (b) the cumulative hydrometeor mass concentrations from each $V_{dop}$-altitude bin. Red contours represent "cloud (CL)", green contours represent "rain (RN)" blue contours represent "ice and snow (IS)", and purple contours represent "graupel and hail (GH)".



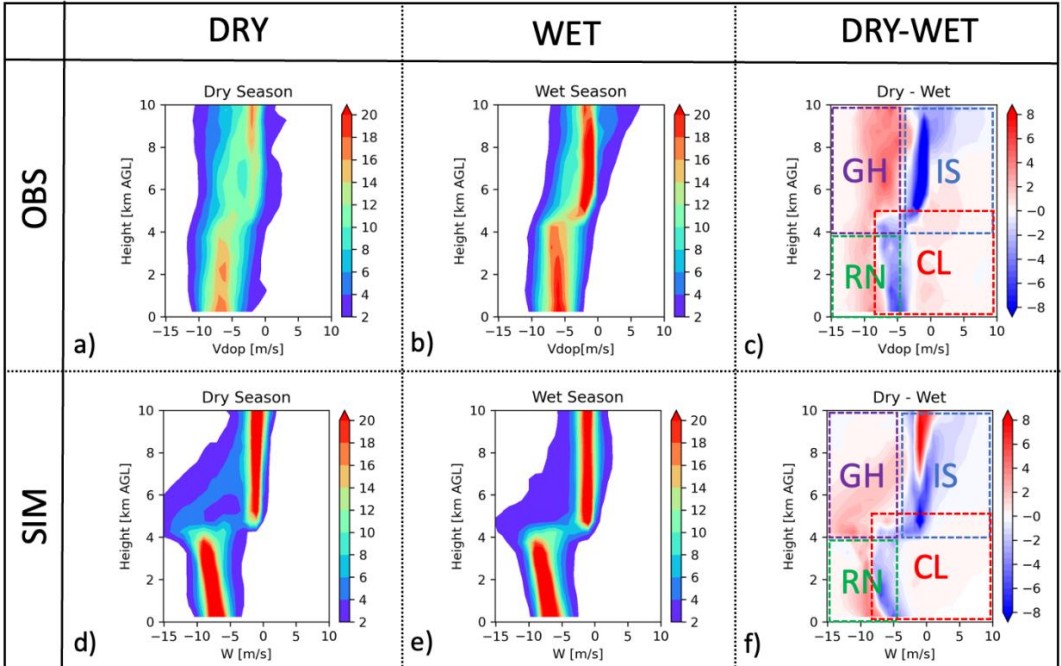

Figure 4. Contoured Frequency of Altitude Diagram (CFADs) of $V_{dop}$ integrated over dry and wet periods, as well as dry-wet differences. The upper raw (a-c) represents observed composites, while the lower raw represents simulated composites. CL, RN, IS, and GH represent the hydrometeor regimes defined in Fig. 3.



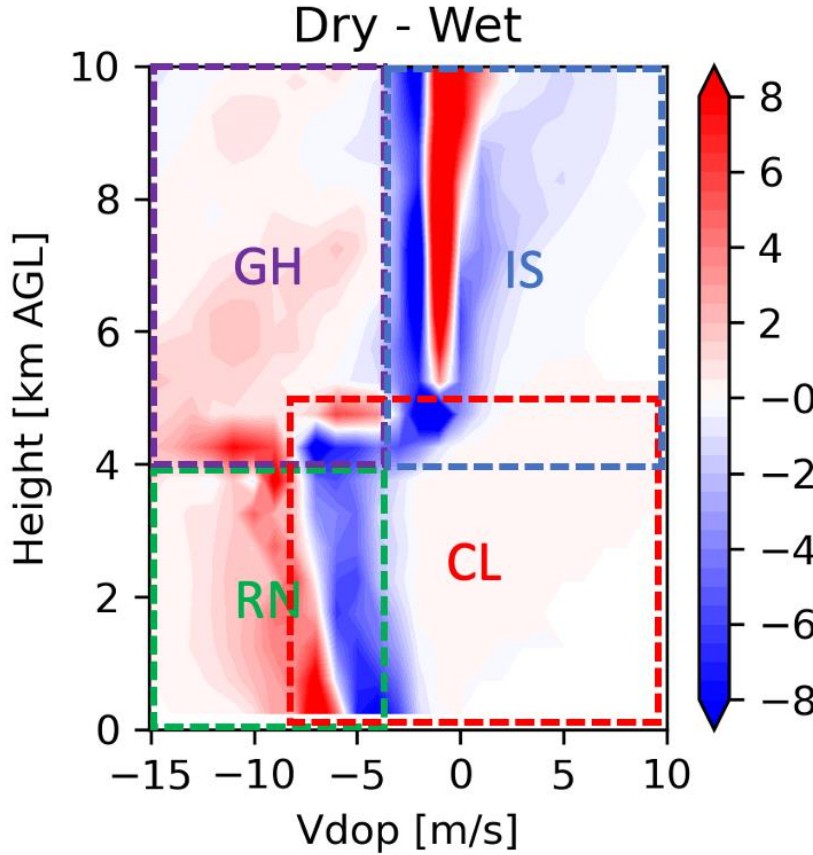


Figure 5. Contoured Frequency of Altitude Diagram (CFADs) of simulated $V_{dop}$, differentiated for
dry- and wet-season golden cases.




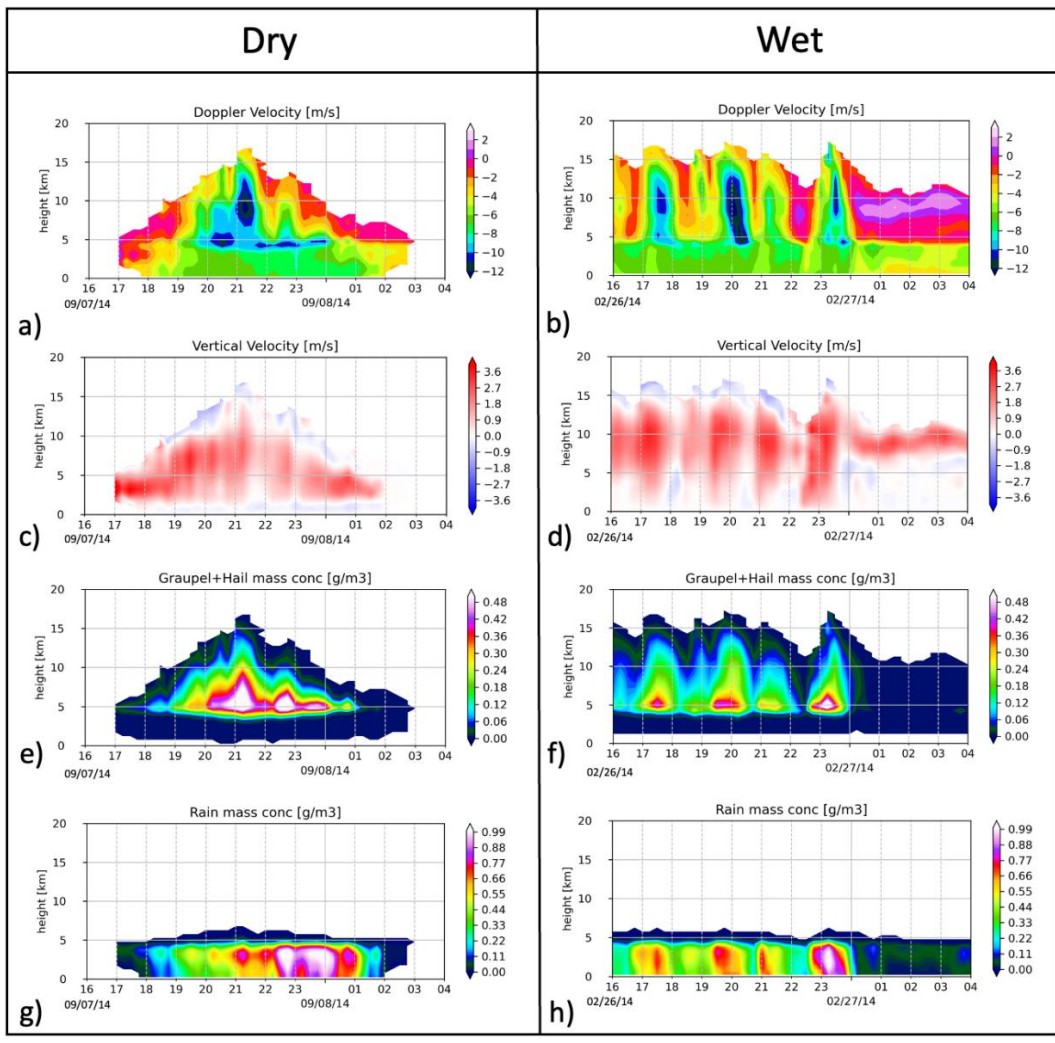

Figure 6. Time series of domain-mean (a-b) Doppler velocity, (c-d) vertical velocity, (e-f) graupel and hail concentrations, and (g-h) rain concentrations profiles of convective grids from the dry- and wet-season golden cases.

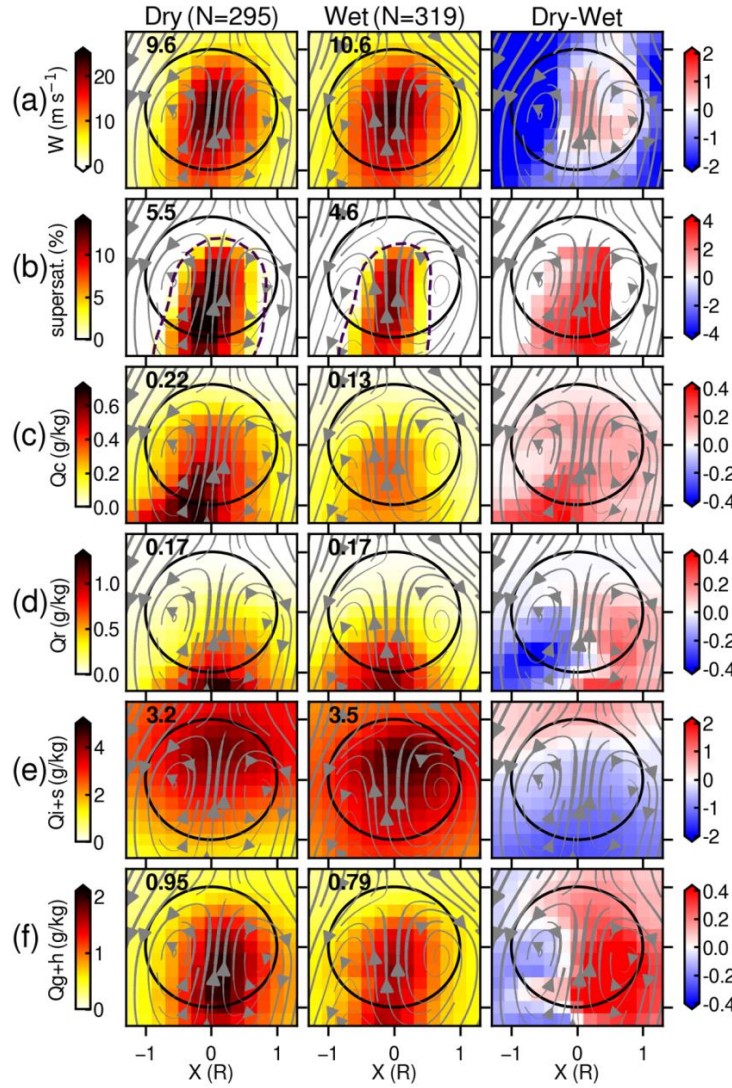

Figure 7. Cross sections along the x-z plane of flux-weighted thermal values of (a) vertical velocity (W), (b) supersaturation (S), (c) cloud droplet mass concentration (Qc), (d) rain mass concentration (Qr), (e) ice and snow mass concentration (Qi+s), and (f) graupel and hail mass concentration (Qg+h), for composites of all tracked thermals at the point of their maximum vertical velocity, scaled by their radius (horizontal and vertical coordinates are in units of mean thermal radii). Left, middle, and right column corresponds to the dry-season golden case, the wet-season golden case, and dry-wet case difference, respectively. Upper left values in each panel are the flux- and radius-weighted mean over all samples. Arrows indicate the average flow streamlines in the rising thermal reference frame. The dashed contour in supersaturation values corresponds to 100% relative humidity. These are reference-time (t=0) mean values.

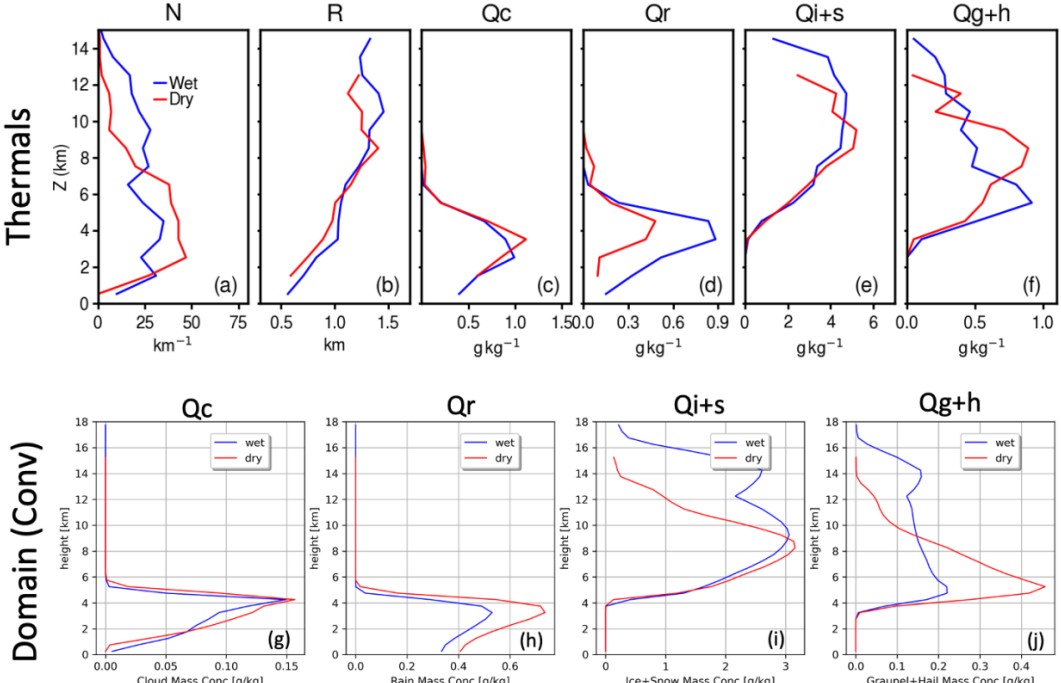

Figure 8. (a-f) Vertical profiles of thermal-mean (a) number concentrations, (b) radius, (c) cloud droplet mass concentration, (d) rain mass concentration (Qr), (e) ice and snow mass concentration (Qi+s), and (f) graupel and hail mass concentration (Qg+h). These are all-thermal mean values. (g-j) Vertical profile of domain-mean (g) cloud droplet mass concentration (Qc), (h) rain mass concentration (Qr), (i) ice and snow mass concentration (Qi+s), and (j) graupel and hail mass concentration (Qg+h) of convective grids from the dry- and wet-season golden cases.



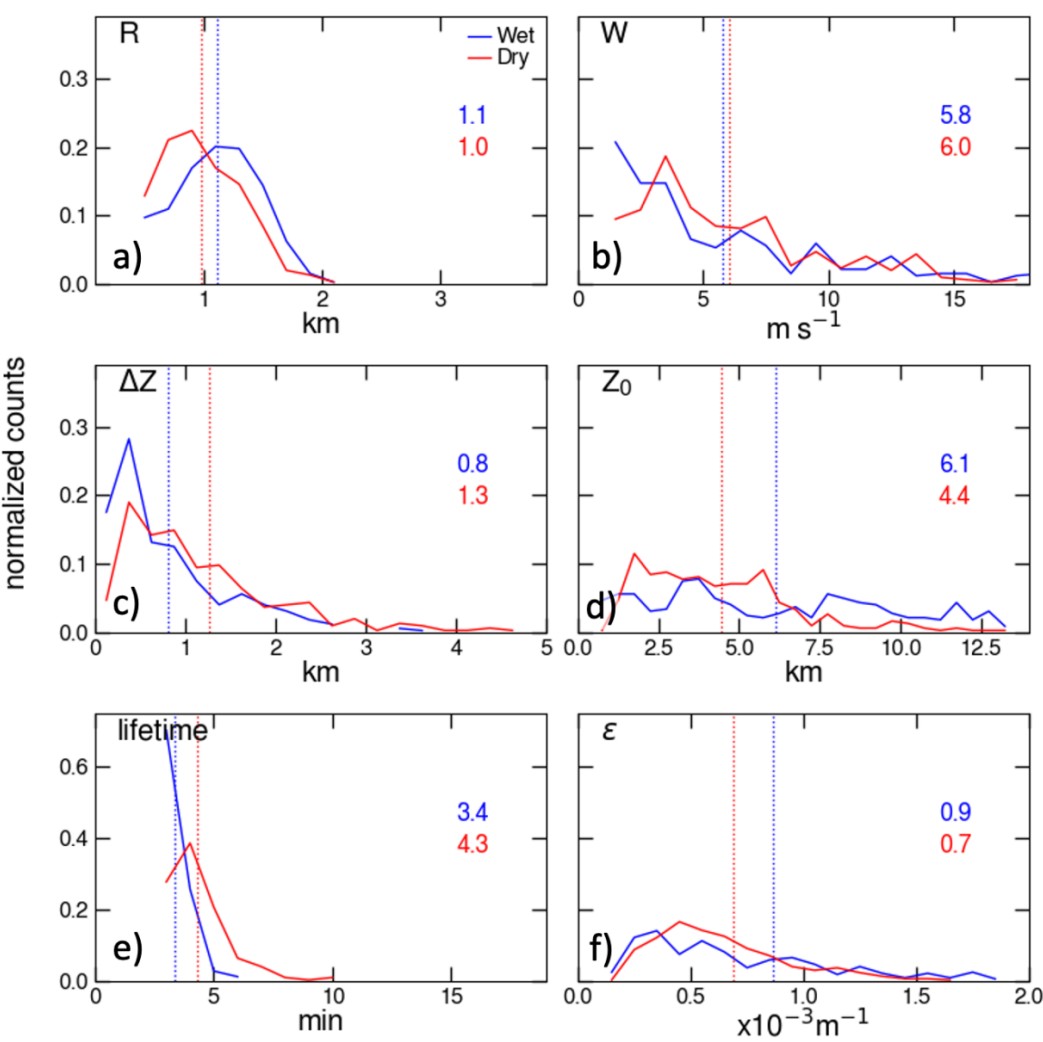

Figure 9. Normalized histogram of thermal (a) radii, (b) vertical velocity (W), (c) travel distance (DZ), (d) initiated level ($Z_0$), (e) lifetime, and (f) entrainment rate (e) from the dry- and wet-season golden cases.



Figure 10. Time series of cross sections along the x-z plane of thermal mean values of vertical velocity, supersaturation values, cloud droplet mass concentration (Qc), rain mass concentration (Qr), (e) ice and snow mass concentration (Qi+s), and graupel and hail mass concentration (Qg+h), for composites of all tracked thermals scaled by their radius.






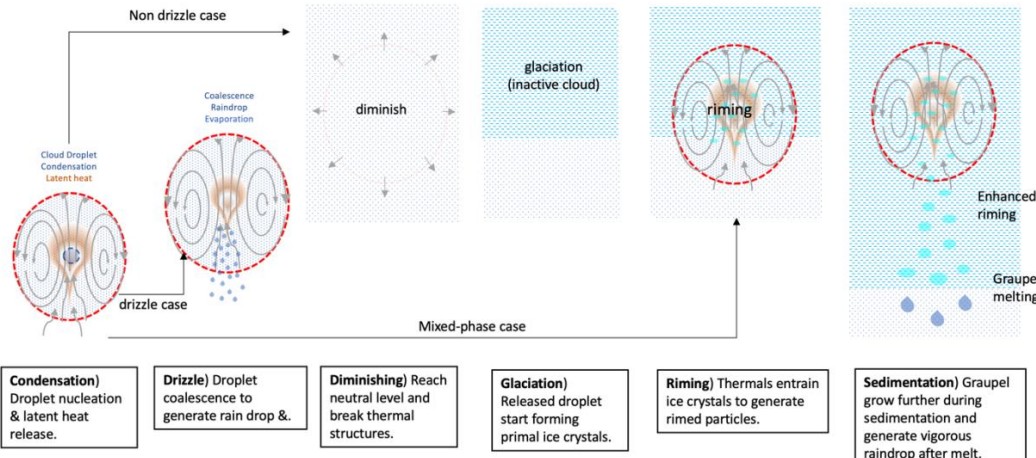


a)

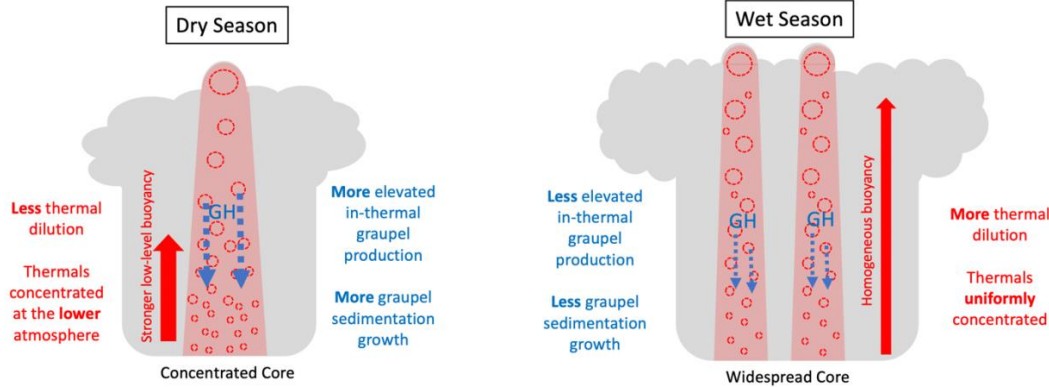


b)
Figure 11. (a) Diagram of the suggested mechanisms for generating graupel and hail through
thermal processes. (b) Diagram of thermal characteristics in deep convection in the dry and wet
seasons.