# Peer review of "Thermal-Driven Graupel Generation Process to Explain Dry-Season Convective Vigor over the Amazon"

_EGUsphere, 2024_

## Referee Comment (RC1)

Review comments for "**Thermal-Driven Graupel Generation Process to Explain Dry-Season Convective Vigor over the Amazon**"

**General comments**

This manuscript conducted a series of LES experiments to examine the dry-wet season contrast on the evolution of deep convection processes, particularly, the interaction between updraft velocity and microphysics on convective vigor. Through tracking thermals in simulated golden cases, it found that dry-case (wet-case) convection tends to generate more (less) numbers of droplet-loaded thermals in the lower atmosphere, corresponding to larger potential buoyant energy at a low level in the dry season. Greater concentration of low-level thermals can potentially lead to more cold precipitation production and convection invigoration. In general, this manuscript is well designed, which uses both LES simulation and observations to investigate underlying processes. It is also quite novel to track microphysical properties within cloud thermals rather than updraft plumes, which provides a new perspective to understand updraft-microphysics interaction. Overall, I think this manuscript presents great work that is worthy of publication. I just have some minor comments as follows:

**Minor comments:**

1. L165: "without expectations for larger hail in Amazon deep convective storms", is there any reference or literature to support this statement?

2. L221: What about the initial condition and surface fluxes? How to initialize each run? I don't see any description of these conditions.

3. L429: There also seems to be discrepancy in GH and RN categories as well. Did you compare the simple statistics such as precipitation and cloud fraction between observation and simulation for more direct comparisons? This comparison can directly reveal how well the simulated case represents typical cloud and precipitation environment of each season.

4. L482: What is the definition for "golden events"? I don't seem to find this in the manuscript.

5. L601: Could you elaborate on this statement? If the thermal size is smaller, shouldn't we expect more rather than less dilution from the environment?

6. L607: Again how so?

7. L619: lesser -- fewer

8. L622: normalizing -- normalization

---

## Author Comment (AC1)

*General comments*:

*This manuscript conducted a series of LES experiments to examine the dry-wet season contrast on the evolution of deep convection processes, particularly, the interaction between updraft velocity and microphysics on convective vigor. Through tracking thermals in simulated golden cases, it found that dry-case (wet-case) convection tends to generate more (less) numbers of droplet-loaded thermals in the lower atmosphere, corresponding to larger potential buoyant energy at a low level in the dry season. Greater concentration of low-level thermals can potentially lead to more cold precipitation production and convection invigoration. In general, this manuscript is well designed, which uses both LES simulation and observations to investigate underlying processes. It is also quite novel to track microphysical properties within cloud thermals rather than updraft plumes, which provides a new perspective to understand updraft-microphysics interaction. Overall, I think this manuscript presents great work that is worthy of publication. I just have some minor comments as follows:*

*Minor comments*:

*1. L165: "without expectations for larger hail in Amazon deep convective storms", is there any reference or literature to support this statement?*

We added the following citations for this statement.

Liu, C., & Zipser, E. J. (2015). The global distribution of largest, deepest, and most intense precipitation systems. *Geophysical Research Letters*, *42*(9), 3591-3595. https://doi.org/10.1002/2015GL063776

Bang, S. D., and D. J. Cecil, 2019: Constructing a Multifrequency Passive Microwave Hail Retrieval and Climatology in the GPM Domain. J. Appl. Meteor. Climatol., 58, 1889–1904, https://doi.org/10.1175/JAMC-D-19-0042.1.

*2. L221: What about the initial condition and surface fluxes? How to initialize each run? I don't see any description of these conditions.*

A large-scale forcing (LSF) analysis is used to initialize the GCE atmospheric profile and also to drive GCE through forcing nudging (e.g., Matsui et al. 2023).

We added the following sentences.

"Each daily LES is initialized at 12 AM local time **from the LSF** and"

"The horizontal/vertical advective forcings, mean wind profiles, and surface heat fluxes from the GoAmazon LSF are used to drive the LES. These fields are interpolated and imposed as tendency terms at every model time step."

*3. L429: There also seems to be discrepancy in GH and RN categories as well. Did you compare the simple statistics such as precipitation and cloud fraction between observation and simulation for more direct comparisons? This comparison can directly reveal how well the simulated case represents typical cloud and precipitation environment of each season.*

We previously mentioned the GH discrepancy in the preceding paragraph. The slight biases in the RN category should be connected to the GH biases. We did not want to emphasize these biases too much in this paper. Instead, we chose to focus on the performance in the dry-wet difference.

Our previous paper (Tao et al. 2022) has already validated precipitation, and the results showed that GCE follows the observed (forcing precipitation) quite well when forced by the LSF. However, we did not conduct any cloud fraction validation since GCE uses a doubly cyclic boundary condition with narrow domains where advected upper-level clouds often dominate the cloud fraction.

We have added the following citation and sentence in Section 2.2:

"Our previous research demonstrated that GCE reproduced the observed precipitation quite well when forced by the GoAmazon LSF (Tao et al. 2022)."

Tao, K., Iguchi, T., Lang, S., Li, X., Mohr, K., Matsui, T., & Braun, S. (2022). Relating Vertical Velocity and Cloud/Precipitation Properties: A Numerical Cloud Ensemble Modeling Study of Tropical Convection. *Journal of Advances in Modeling Earth Systems*, *14*(9), e2021MS002677. https://doi.org/10.1029/2021MS002677

*4. L482: What is the definition for "golden events"? I don't seem to find this in the manuscript.*

*For clarification, we have modified the following sentence from*
*"Namely, we select two single-day simulation cases representing typical dry and wet-season convection."*
*to*
"Golden cases are two single-day simulation cases, one for dry and one for wet season, representing $V_{dop}$ CFADs of each season."

*5. L601: Could you elaborate on this statement? If the thermal size is smaller, shouldn't we expect more rather than less dilution from the environment?*

Our results did not support our typical assumption that smaller thermals tend to have more dilution. Our previous research, conducted by Hernandez-Deckers and Sherwood in 2018, also indicates that the variability of thermal radius can only account for 20% of the total variance of thermal dilution rate. This was concluded through similar thermal tracking analysis from different LESs. Therefore, thermal size cannot be considered the sole predominant parameter for determining dilution rate.

We added the following new sentence.

"Interestingly, slightly smaller thermal radii in dry-case convection can have a lower entrainment rate than in wet-case convection. Hernandez-Deckers and Sherwood (2018) also found that the variability of thermal radius can only account for 20% of the total variance of thermal dilution rate, i.e., larger thermal tends to have a lesser dilution rate. This was concluded through similar thermal tracking analysis from different LESs. Therefore, thermal size is not the sole parameter for determining dilution rate."

*6. L607: Again how so?*

It is a good question. It's speculated that the dry season convection needs more energy to break a stable atmosphere when the shallow-to-deep transition occurs in convection. This process will likely concentrate updraft energy in a narrow area because most of the atmosphere remains stable in the dry season. This will be subject to future investigation.

We added the following sentence.
"This is merely speculative and requires further investigation to confirm or refute."

*7. L619: lesser -- fewer*

Corrected.

*8. L622: normalizing – normalization*

Corrected.

---

## Author Comment (AC3)

*General comments:*

*This manuscript uses the large eddy simulations to study moist convection over Amazon in dry and wet seasons and compared to field observations during Go-Amazon. The unique results are from thermal analysis implying the microphysical processes throughout the development of thermals. This work is very valuable in understanding the moist convection process and should be shared with the community. However, some discussions are still in need of clarification. Therefore, I recommend accepting the manuscript with some minor-major revisions.*

Thanks for your valuable input. Our response and revisions are discussed below.

*Major comments:*

*Though there are a lot of great results in this manuscript, the statement of the major ice growth by riming on entrained ice particles into the thermal from top is not well supported by the figures. Figure 10 shows that ice seems coming from top of thermal. But it includes the effect of thermal growth and move upward into colder temperature and activate ice from top while it looks like ice falling into the thermal. This part should be carefully reexamined with environment temperature as a reference, similar like Figure 10, but using temperature range instead of lifetime.*

It is important to consider the lifecycle reference due to the differences in thermal vertical velocity strength at different stages. However, as you suggested, it is also useful to examine thermals at reference temperature. To make things easier for our algorithm, we have created thermal composites with altitude as the reference point. We have included new Figure 11 for these composites, along with additional discussion, in Section 4 of our manuscript. Bottom line is that the new plot augments our findings and does not change the conclusion.

"In addition to reference time, we composited thermal properties at different altitude levels (Fig. 11). The method for sampling and compositing is the same as in Fig. 10. However, it characterizes vertical profiles of thermal composites using altitude references from 2.5km up to 10.5km. Despite of different reference methods, altitude-reference plots appear to be similar patterns to time-reference plots. Vertical velocity (W) increases toward the peak level at 8.5 km, similar to the mean profile in Fig. 8c. There is strong supersaturation (S) between 4.5 km and 6.5 km, which rapidly decreases above 8.5 km. Qc, Qr, Qi+s, and Qg+h profiles also resemble thermal-mean profiles in Figs. 8d, 8e, 8f, and 8g, respectively. In thermal cores, Qc, Qr, and Qg+h are initially concentrated but sediment as thermals ascend. Even compared with the lifetime composite in Fig. 10, Qi+s are more stratified horizontally rather than toward thermal cores, particularly from 3.5 km to 6.5 km, suggesting entrainment of Qi+s within thermals at these altitudes. Simultaneously, Qc and Qr decrease while Qg+h increases at these altitudes. Because wet growth (ice collecting supercooled liquid) is the primary graupel growth process within 4ICE microphysics in the GCE

(Lang et al. 2014), the droplet-loaded thermal penetration toward ice layers appears to be the important graupel/hail generation process within tropical deep convection. "

*Normalizing through the life cycle of thermals may not be the best approach because it removes the dependence of time. Microphysics process is quite time sensitive, e.g. warm rain process, riming process. The time and path for hydrometeor growth is the key to interpret the differences in dry and wet moist convection. Current manuscript does discuss a little on this with Figure 9 (L592-602). But more can be added, such as velocity vs altitude in Figure 8, or the tracking thermal time at different temperatures.*

Please note that our lifecycle plot (Fig 10) updates every minute and does not normalize for varying lifetimes. Thus, it can still detect time-dependent processes in Fig 10.

*Maybe add some discussion on the limitations of how this result could apply globally. For example, dry season Amazon compared to the central plain of US, or Argentina.*

This tracking for mid-latitude organized convection with much horizontal wind shear has not been extensively tested yet. We have added the following paragraph in Section 4.

"It is well known that severe hailstorms and large hails are more frequently observed over the central plains of North and South America (Liu and Zipser 2015, Bang and Cecil 2019). The hailstorms in these regions are often associated with supercells, and mesocyclones play a crucial role in the growth of very large hail by enhancing the residence time of the hails within the mixed-phase zone (e.g., Nelson 1983; Ziegler et al. 1983). However, it is not known how a mesocyclone affects thermal-like or plume-like updraft characteristics (Morrison et al., 2020), though satellites have captured thermal chain-like periodic overshooting signals at the top of the supercells (Borque et al. 2020). Thus, further observational and modeling investigations are required for the mid-latitude regions to determine whether our proposed graupel/hail generation mechanisms can be applied."

Borque, P., Vidal, L., Rugna, M., Lang, T. J., Nicora, M. G., & Nesbitt, S. W. (2020). Distinctive Signals in 1-min Observations of Overshooting Tops and Lightning Activity in a Severe Supercell Thunderstorm. *Journal of Geophysical Research: Atmospheres*, *125*(20), e2020JD032856. https://doi.org/10.1029/2020JD032856

Nelson, S. P., 1983: The influence of storm flow structure on hail growth. *J. Atmos. Sci.*, 40, 1965–1983, https://doi.org/10.1175/1520-0469(1983)040<1965:TIOSFS>2.0.CO;2.

Ziegler, C. L., P. S. Ray, and N. C. Knight, 1983: Hail growth in an Oklahoma multicell storm. *J. Atmos. Sci.*, 40, 1768–1791, https://doi.org/10.1175/1520-0469(1983)040<1768:HGIAOM>2.0.CO;2.

*Minor comments:*

*L61, radiation warms temperature? Maybe surface and increase skin temperature*
Corrected.

*L81, what do you mean by "vigorous raindrops"?*
Corrected to "large and copious".

*L108, please be specific on stronger precipitation properties, e.g. larger raindrops, more raindrops, more cloud droplets, larger reflectivity?*
"stronger precipitation properties" is replaced by "larger reflectivity".

*L140, replace "collected by" with "of"*
Corrected.

*L170, I do not believe that you can calibrate the whole profile of radar measurements with surface disdrometer. Should this be 'all radar measurements at near surface were calibrated…'*

The RWP collects time-height profiles of signal-to-noise ratio (SNR) which maps to reflectivity factor Z. "Calibrating" the RWP against a surface disdrometer reference implies establishing a fixed calibration offset "C" for estimates of Z following a standard radar equation: $Z(r) = SNR(r) + 20\log(r) + 'C'$ [dBZ], where 'r' is range/height for a profiling radar. Here, low-level rain properties act as reference, most often from widespread stratiform rain well-captured by the disdrometer. For examples of such procedures and considerations, see Williams et al. (2023). The radar "C" that best matches with the disdrometer applies to the entire column. For example, if the radar was instead calibrated by comparing Z offset from a nearby NEXRAD versus RWP at a higher altitude, the associated constant "C" that best matched would be expected to be the same provided that NEXRAD was also well-calibrated. As the RWP is a longer wavelength radar, it is attenuated in rain or from gaseous attenuation.

This may be too much information beyond the scope of our manuscript, potentially disrupting readers. So we very briefly modified (red part) the sentence as follows.

"All radar measurements were calibrated against a reference laser disdrometer collocated at the main AMF site during the campaign (e.g., Wang et al. 2018), and a detailed method is discussed in Williams et al. (2023)."

Williams, C. R., Barrio, J., Johnston, P. E., Muradyan, P., and Giangrande, S. E.: Calibrating radar wind profiler reflectivity factor using surface disdrometer observations, Atmos. Meas. Tech., 16, 2381–2398, https://doi.org/10.5194/amt-16-2381-2023, 2023.

*L216, now everyone is using ERA5, but I do not expect there is a big difference in the forcing if interim is used. But it is still interesting to know how small the difference is just for one case.*

Unfortunately, we did not test the forcing from the interim because of forcing data availability. DOE team uses ERA5 to construct large-scale forcing. But we believe there's some sensitivity due to the improvement of ERA5 over the interim, but the investigation is beyond the scope of this manuscript.

*L269, can thermals merge? This 80% criteria could be problematic if somehow there is a merge if model resolution is not high enough.*

The tracking algorithm we use here cannot account for thermal merging or splitting. To account for this, one would have to modify this 80% threshold, indeed. However, this is beyond the scope of this work. Let us add the following sentence for future study in Section 4.

"Note that our tracking algorithm cannot analyze merging or splitting thermals, regardless of their occurrence. This limitation requires further study."

*L278, what is the discard rate in your study, especially for those long-lasting ones? I am wondering how many thermal samples would not be follow the momentum budget, and why.*
The discard rate of thermals of our tracking algorithm was discussed in detail by Hernandez-Deckers and Sherwood (2016) (see section 3c there, and table 2). It is important to note that thermals are not directly discarded "as a whole", but rather individual tracked time steps may be removed if there is a momentum budget mismatch: "If the mismatch is too large, we remove time steps from the beginning or the end of the tracked period—whichever has lower updraft speed—until the mismatch is acceptable or until we discard the entire thermal" (Hernandez-Deckers and Sherwood, 2016).

We found that the "momentum budget mismatch" is not the main reason for completely discarding thermals (consistent with Hernandez-Deckers and Sherwood, 2016). Only 3% of the thermals in the dry case and 8% in the wet case are entirely discarded due to this. However, in terms of discarding individual time steps, 19% of the identified time steps are discarded due to momentum budget mismatches in the dry case, and 28% in the wet case. This shortens the tracked lifetime of thermals but does not necessarily discard the thermals completely. As discussed by Hernandez-Deckers and Sherwood (2016), momentum budget mismatches tend to occur during the initial or final stages of the thermals, when they are less organized and less vigorous. Slow, less-buoyant

thermals may deviate more from the idealized spherical shape that the tracking algorithm assumes, which we hypothesize to be the main reason for the momentum budget mismatch. The fact that more time steps are discarded for this reason in the wet case is consistent with the fact that vertical velocity tends to be slower there compared to the dry case.

*L385-386, I am confused on this. Do you mean that terminal velocity is not included in the velocity shown in Figure 3? Then it should not be called Vdop. Please clarify.*
$V_{dop}$ is the sum of the vertical velocity of the air and the terminal velocity of hydrometeor particles weighted by their reflectivity. However, in the GCE, the terminal velocity of cloud droplets is zero, which is why the variability of Vdop in the CL regime mainly reflects the vertical velocity of the air. That's what we meant in this sentence. We added the following sentence in this paragraph.

"Because $V_{dop}$ is the sum of the vertical velocity of the air and the terminal velocity of hydrometeor particles weighted by their reflectivity, this plot facilitates understanding $V_{dop}$ CFAD. "

*Figure 3, font of xytitle is too small to read.*
Corrected.

*Figure 3 caption, What kind of samples are included? Any sample with certain Q? Convective region? Please clarify.*
The convective core, defined at the end of Section 2.1, samples this. We added "convective core" in the Fig. 3 titles.

*L398-401, another important reason is that model overestimates the graupel and hail too fast at warmer temperatures, which leads to larger melted rain drops and high terminal velocity.*
We added the following sentence.

"Another possible reason is overestimating the graupel/hail size, leading to larger melted raindrops with high terminal velocity."

*Figure 6, caption, "domain mean" (include rain or no rain) or domain mean of convective cores?*
Yes, it is a convective core. Let us change it to "convective-core mean".

*Figure 7. Have you tried to put streamline (or vector differences) of dry-wet in right panels?*
No, unfortunately. The plot will be quite messy. However, we have included quantifying mean velocity and entrainment rate in Fig. 9.

*L552-553, the boundary layer eddies and the moist convective thermals are two different concepts and should not be mixed here. Larger boundary layer eddy could lead to a stronger initial velocity at cloud base. It should have minimum relationship to the initial thermal size.*

We have included Figure 7 from William and Stanhill's 2002 study below. The figure illustrates the relationship between the height of boundary layers and the width of dry and moist thermals. It also shows the transition from dry to moist thermals, which are similar concepts but differ in whether or not the air is saturated and condensation occurs. Some dry thermals that start at the boundary can turn into moist thermals once they reach the lifting condensation level. Morrison et al. (2021) argue that the growth rates of moist thermals are smaller due to latent heat release and different buoyancy distributions.

[Figure]

**Figure 7.** Illustration that thermal width and moist convective updraft width scale with the height of cloud base. The scale-independent angle of plume expansion in the vertical is based on similarity theory and supported by laboratory scale experiments with buoyant plumes.

Our study does not support the argument from William and Stanhill (2002), that a deeper boundary layer tends to generate larger thermals. Instead, we found that stronger convective cores are generally characterized by many cumulus thermals in terms of their number rather than their size. Thermal speeds are also faster at mid-level in dry-case convection, too. Nevertheless, it would be beneficial to examine the transition from dry to moist thermals in future studies that aim to investigate both the PBL and convection seamlessly.

We added red words in the sentence.
"the width of dry/moist thermals should increase with increases in the boundary layer depth"

We also added the below sentence in Section 4.
"For this, we need to investigate boundary layer dry thermals and deep-convective moist thermal seamlessly between various continental and maritime environments."

*L558, differences not discussed here include the water load and where thermal starts. 8b shows that thermals start with the same size and growing up for dry and wet cases the same way, except dry starts at a higher altitude. Wet has more cloud water at 3 km due to longer path and weaker vertical velocity leading to a longer time for warm rain process. Dry has stronger vertical velocity to start and a shorter path so most of cloud droplets reach freezing level (8c) to help growing graupel and hail. Note that dry supposes to have higher vertical velocity at the beginning.*

According to new Fig. 8c, updraft velocity of thermal below 4km are nearly identical between dry and wet cases. However, dry-case thermals peaks updraft velocity at 8km altitude, coincided with Qg+h peak. At least, thermal sizes are not a key factor that characterizes the dry/wet case deep convection over Amazon. We have modified the sentence.

"Further, Fig. 8c shows the mean vertical velocity of thermals, showing nearly identical updraft velocity below 4 km. However, mean updraft velocity profiles above 4 km are quite different between the dry and wet cases; e.g., the dry case peaks around 8 km, while the wet case peaks at 6 and 10 km. This result implies that differences in moist convection between dry and wet cases are poorly characterized by thermal sizes. "

*Figure 8. I am very surprised that the vertical velocity is not included in Figure 8. This is almost the most important variable for whole study! Please add that in the revision.*

Mean vertical velocity is added in Figure 8, and related discussion (above).

*L606, can "have"*
*This is OK.*

*Figure 9. R, W, and Entrainments are all a function of altitude/temperature. It would be great if CFADs can be made for them. For life time, it would be great if the proportion of time in > 0C can be added.*

That's a nice suggestion, but unfortunately, we won't have enough samples to carry it out in this study. As you can see in Fig. 9, it's already quite spiky even when all altitudes are lumped together. If we were able to generate thermal tracking data from entire seasons, it would be possible to do, but it would be quite challenging due to the computational cost. We added the following sentence.

"Nevertheless, these parameters also depend on altitude. More thermal samples can allow us to characterize vertical profiles of histograms in a future study."

*Figure 10, Caption or y title. what is the unit of time lag here? What kind of thermal samples are included? Please clarify.*

All cumulus thermals were sampled in the calculations. X and Y-axis are normalized thermal size. The units of time lag is in minute (added in the Figure captions), and we have already provided details of the calculation method below.

*"For this, each thermal at its maximum w value is centered and defined as the time of zero, and prior (later) steps are represented in negative (positive) time steps. Because of the 1-minute LES output, the time series from -3 to 3 encompasses 7 minutes of time steps. This averaging process also weighs upon the magnitude of the thermal mass flux (Hernandez-Deckers et al. 2022); thus, thermals at larger values in positive and negative time steps tend to have a fewer sampling numbers. Also, to make the composites, equal-sized thermals are sampled to characterize the mean time series of thermal properties, avoiding sampling too small thermals, which often has no supersaturation (Hernandez-Deckers and Sherwood. 2016). This normalization procedure ends up with the result that maximum W values do not appear at reference time (t=0), but better capture the evolution of the largest flux-bearing thermals (Fig. 10)."*

*L638-639, I disagree with this statement. If this statement is true, the ice would activate most outside of thermals. Then the follow up question would be why ice nuclei would not activate inside the thermal when it first encounters the cold temperature during rising and expanding. The composite includes thermals centers with upward motion. This could be just thermal transitioning from liquid to ice phase in growing/rising up. At least from this figure alone, this statement is not fully supported.*

We mentioned that ice is not actively generated from thermal at the beginning of the stage when entering the mixed-phase zone. Immediate (i.e., homogeneous) freezing only occurs below -39degC (i.e., homogeneous freezing). In mixed phase zone (0~39degC), heterogeneous ice nucleation and deposition growth are also slower compared with rapid droplet nucleation and condensation growth at moisture-rich lower troposphere (Pruppacher and Klett 1997). If this is purely a transition from liquid to ice as you suspect, $Q_{i+s}$ must be concentrated around the center of thermals, where $Q_c$ is most concentrated. However, $Q_{i+s}$ is much more homogeneously distributed at the beginning.

In the later stages, when most thermals are cold enough and the supersaturation rate drops, the Bergeron process could enhance the ice growth process within the thermal as droplets evaporate. Neverthless. we discussed that ice nucleation remains the largest uncertainty in Section 4. But the bottom line, it is clear that the thermal is being submerged into an ice layer at the beginning of time steps, and graupe/hail are generated at the same time, as indicated in the figure. Thus we have modified the sentence in the following way (red is added).

"This evolution of $Q_{i+s}$ suggests that thermals are not the main initiator of $Q_{i+s}$ at the beginning," "$Q_{i+s}$ becomes a leading destination of the overall transfer from vapor to hydrometeor phases within thermals that remain vigorous probably due to the Bergeron process (Bergeron 1935)."

We removed the following sentence.

"This also suggests that the glaciation process (i.e., conversion from supercooled liquid to ice hydrometeors) is usually completed after thermals vanish unless they reach the upper level of convective cores."

And added the following sentence.

"Further research is needed to investigate the ice nucleation process in greater detail throughout the different stages of thermals' lifecycle. A more detailed analysis of individual thermal pathways as a function of thermal temperature range is likely required."

Bergeron, T., 1935: On the physics of clouds and precipitation. *Proces Verbaux de l'Association de Météorologie,* International Union of Geodesy and Geophysics, 156–178.

*L650, this could just be colder temperature leads to more riming process.*

Enhanced riming requires the collision of supercooled water (wet growth) in GCE microphysics schemes. Colder temperature can greatly ehnahce ice-to-ice collection, but its effect on ice-to-droplet collection (wet growth) is small or negligible. We added the following sentence.

"Note that the 4ICE scheme only allows wet growth (collision against supercooled liquid) of graupel, and dry growth (collision against ice) is suppressed (Lang et al. 2014)."

*L655, though I somewhat agree with the statement, but the evidence shown in the manuscript does not directly disapprove the tradition concept model. Note that Arakawa and Schubert were using models with much coarser resolutions and have no capability to resolve "thermals", which has a very loose definition throughout last half century.*

One sentence is not enough to discuss the details of the plume vs. bubble concept. We ditched the sentence.

"; i.e., a large single air mass lifted from the cloud base up to the cloud top can generate latent heat and precipitation (Arakawa and Schubert 1974)"

*L678-679, Figure 11, if the riming is mainly from entrainment of ice, you would see the enhanced Qi from side edge of thermals in Figure 10.*
Since the droplets are strongly concentrated at the core of the thermals rather than at the side edges, we expect significant riming to be concentrated at the thermal cores. This is discussed in Section 3.3 related to Figure 10.

*L705, 0.8m/s as the definition of thermal could exclude weaker thermals in wet season. At least a discussion on the definition of thermals should be included here for discussion. What if you use 0.1 m/s, do we still see more thermals in dry season? Also, is this only from a golden case day? Or the whole time period? How representative is this to the whole dry and wet season?*

The cumulus thermals investigated in this study are "organized updrafts" in deep convection. 0.1m/s of very weak updraft won't cause any significant supersaturation in this environment. Earlier studies by Hernandez-Deckers and Sherwood (2016) investigated the sensitivity of these thresholds but did not find any significant findings. Even in the previous intercomparison study of updraft velocity, 1 m/s was the threshold to define updraft grids (Marinescu et al. 2022). The 0.1m/s threshold will be appropriate for stratocumulus or equivalent weak cloud systems.

We compared the single-day $V_{dop}$ CFADs to the seasonal $V_{dop}$ CFAD and selected the golden case (Fig 5). This is why we believe that single-day convection reasonably represents the seasonal behavior of convection. Of course, more cases would be better if we had more computational resources. We added the following sentence.

"These new processes are proposed from the "golden" cases from the dry and wet seasons. Further investigation with more case studies will augment our hypothesis in the future."

*L716, this may not be fair to compare to L-O differences. The diurnal variations of land vs. ocean have one with thermals only in part of day and another spread the whole day.*
Well, apart from diurnal variation, the L-O difference can be attributed to other complex phenomena, such as mesoscale circulations (island effect) and aerosol impacts as well. So, we agree that it may not be stated simply here. Instead of adding a new paragraph, let's remove the statement.

The "hotter" surface in the dry season must be the physical origin of step 1.

*L728, it can be either numbers or stronger initial velocities satisfying 0.8m/s criteria. Note that larger eddies may only mean stronger initial velocity to start the thermal.*

It was mid-level thermal speed as indicated in new Fig 8.
We added the words.

"and updraft velocity profile" after the number.

---

## Author Response (AR2)

**Dear Editor,**

**Thanks for your final correction.**

**Toshi**

**Public justification (visible to the public if the article is accepted and published)**:
The authors have thoroughly addressed all comments raised by two reviewers, and I therefore recommend it publication after technical corrections:

line 569: "wet-case thermals around 9km (Fig. 7b)": Fig 7b --> Fig. 8b?

Corrected.

line 674, entrapment: should this be "entrainment"?

Corrected.